# Low affinity binding sites in an activating CRM mediate negative autoregulation of the *Drosophila* Hox gene *Ultrabithorax*

Rebecca K. Delker[1,2☯], Vikram Ranade[3☯], Ryan Loker[3], Roumen Voutev[1,2], Richard S. Mann[1,2] *

**1** Department of Biochemistry and Molecular Biophysics and Systems Biology, Columbia University, New York, NY, United States of America, **2** Mortimer B. Zuckerman Mind Brain Behavior Institute, Columbia University, New York, NY, United States of America, **3** Department of Genetics, Columbia University, New York, NY, United States of America

☯ These authors contributed equally to this work.
* rsm10@columbia.edu

**Data Availability Statement:** All relevant data are within the manuscript and its Supporting Information files.

## Abstract

Specification of cell identity and the proper functioning of a mature cell depend on precise regulation of gene expression. Both binary ON/OFF regulation of transcription, as well as more fine-tuned control of transcription levels in the ON state, are required to define cell types. The *Drosophila melanogaster* Hox gene, *Ultrabithorax* (*Ubx*), exhibits both of these modes of control during development. While ON/OFF regulation is needed to specify the fate of the developing wing (*Ubx* OFF) and haltere (*Ubx* ON), the levels of *Ubx* within the haltere differ between compartments along the proximal-distal axis. Here, we identify and molecularly dissect the novel contribution of a previously identified *Ubx* cis-regulatory module (CRM), *anterobithorax* (*abx*), to a negative auto-regulatory loop that decreases *Ubx* expression in the proximal compartment of the haltere as compared to the distal compartment. We find that Ubx, in complex with the known Hox cofactors, Homothorax (Hth) and Extradenticle (Exd), acts through low-affinity Ubx-Exd binding sites to reduce the levels of *Ubx* transcription in the proximal compartment. Importantly, we also reveal that Ubx-Exd-binding site mutations sufficient to result in de-repression of *abx* activity in a transgenic context are not sufficient to de-repress *Ubx* expression when mutated at the endogenous locus, suggesting the presence of multiple mechanisms through which Ubx-mediated repression occurs. Our results underscore the complementary nature of CRM analysis through transgenic reporter assays and genome modification of the endogenous locus; but, they also highlight the increasing need to understand gene regulation within the native context to capture the potential input of multiple genomic elements on gene control.

## Author summary

One of the most fundamental questions in biology is how information encoded in genomic DNA is translated into the diversity of cell-types that exist within a multicellular

**Funding:** National Institute of General Medical Sciences R35GM118336 to Prof. Richard S. Mann, and National Institute of General Medical Sciences R01GM058575 to Prof. Richard S. Mann. The funders had no role in the study design, data collection and analysis, decision to publish, or preparation of the manuscript.

**Competing interests:** The authors have declared that no competing interests exist.

organism, each with the same genome. Regulation at the transcriptional level, mediated through the activity of transcription factors bound to *cis*-regulatory modules (CRMs), plays a key role in this process. While cell types are often characterized by the specific sub-set of genes that are transcriptionally ON or OFF, it is equally important to consider the more fine-tuned transcriptional control of gene expression level. We focus on the regula-tory logic of the Hox developmental regulator, *Ultrabithorax* (*Ubx*), in fruit flies, which exhibits both forms of transcriptional control. While ON/OFF control of *Ubx* is required to define differential appendage fate in the T2 versus T3 thoracic segments, more fine-tuned control of *Ubx* transcription levels is observed in distinct compartments within the T3 appendage. Through genetic analysis of regulatory inputs, and dissection of a *Ubx* CRM in a transgenic context and at the endogenous locus, we reveal a compartment-spe-cific negative autoregulatory loop that dampens *Ubx* transcription to maintain distinct transcriptional levels within a single developing tissue.

## Introduction

Although nearly all cells within a developing animal share the same genome, how genes are deployed, largely by transcriptional regulation, throughout space and time differs dramatically between cells. In eukaryotic cells, transcriptional regulation is governed by the presence of non-protein coding cis-regulatory modules (CRMs) through which the binding of transcrip-tion factors (TFs) can either positively or negatively affect the expression of target genes. While it is common to think of cell identity as a product of a binary ON vs. OFF control of transcrip-tion, an added layer of complexity is gained by regulating the quantitative amount (the level/ dose) of gene expression of both TFs and their downstream targets [1–3]. Here, studying the Hox gene, *Ultrabithorax* (*Ubx*), in *Drosophila melanogaster* (*D. melanogaster*), we explore the question of how gene expression levels can be tuned within the ON transcriptional state.

*Ultrabithorax* (*Ubx*), required during development at both embryonic and larval stages, is best known for its role in the differential development of the second and third thoracic dorsal appendages, the wing (T2) and haltere (T3), respectively [4]. These serially homologous struc-tures are derived from larval imaginal discs that differ primarily in the expression of *Ubx*: a *Ubx* OFF state is required for wing development, while a *Ubx* ON state is required for haltere development [5–8]. While this demonstrates the importance of an ON/OFF binary mode of control for Ubx expression during development, distinct *Ubx* expression levels occur along the proximal distal axis within the haltere imaginal disc (Fig 1A and 1B). While distal cells exhibit high levels of Ubx, proximal cells exhibit low levels of Ubx, resulting in a distal:proximal ratio > 1. This observation raises two questions: (1) what are the downstream consequences of high versus low *Ubx* expression on cell- and tissue-fate within the haltere; and (2) what is the mechanism by which Ubx levels are regulated in proximal versus distal compartments. Here, we address the latter question and provide evidence for the presence of an autoregulatory mechanism in which Ubx directly down-regulates its own expression within the proximal compartment of the developing haltere.

The regulation of *Ubx*, like many developmental regulators that need to be expressed in precise spatio-temporal patterns, relies on input from a number of CRMs throughout its large, ~120 kb locus [7,9–15]. We find a novel role for a previously identified *Ubx* CRM, termed *anterobithorax* (*abx*), located within the large third intron of *Ubx*. Established as an enhancer necessary for activation of *Ubx* within the anterior compartment of the haltere [9,11], we find that *abx* also serves a role in the negative autoregulation of *Ubx* that results in distinct levels of

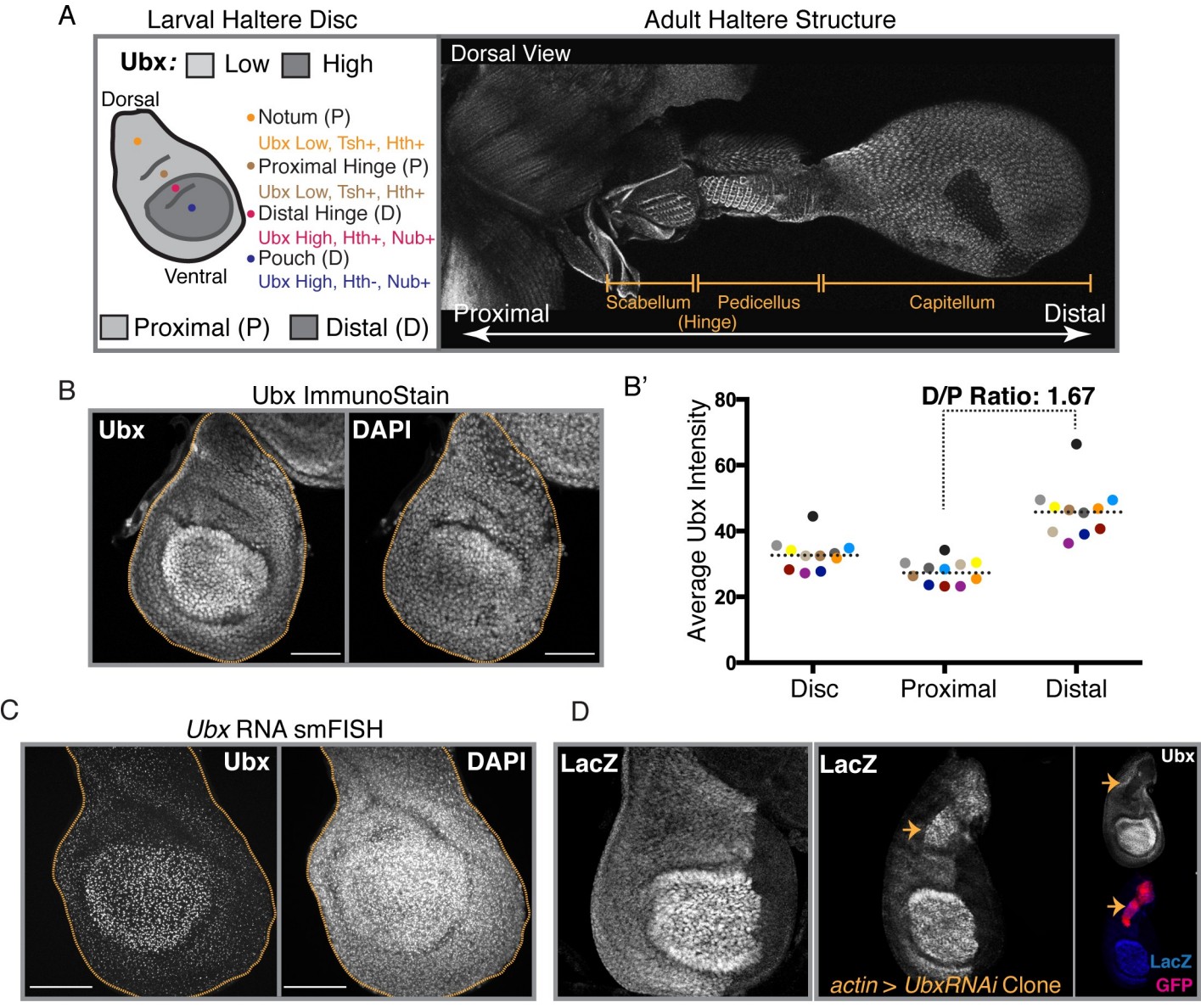

**Fig 1. Proximal:Distal expression bias of *Ubx* is established at the level of transcription.** **(A)** (Left) A schematic of the 3rd Instar haltere imaginal disc, denoting regions of high versus low *Ubx* expression along with regions fated to become the notum (yellow), the proximal hinge (light brown), the distal hinge (pink) and the pouch (blue). Domains of expression of key proximal (*teashirt* (*tsh*), *homothorax* (*hth*)) and distal (*nubbin* (*nub*)) specifying genes are shown. (Right) A confocal image (MAX projection) of an adult haltere. Dorsal view is shown. Structures along the proximal-distal axis are specified. **(B)** Ubx immunostain (along with DAPI nuclear stain) of a 3rd Instar haltere imaginal disc showing proximal-distal expression bias. Haltere disc is outlined in yellow. MAX projection of 4 slices shown. **(B')** Quantification of average Ubx immunostain intensity in the whole disc ("Disc"), proximal compartment, and Distal compartment. Each colored point represents a single disc (N = 11). Dotted line represents the mean for each compartment. Reported D/P ratio is computed by dividing the mean distal intensity by the mean proximal intensity. **(C)** Single-molecule RNA FISH (smRNA FISH) against the first intron of *Ubx* (along with DAPI nuclear stain) showing proximal-distal bias at the level of transcription. MAX projection of all slices shown. **(D)** (Left) LacZ immunostain driven by the enhancer trap insertion, *lacZ^{HC166D}*. (Right) LacZ immunostain driven by *lacZ^{HC166D}* upon generation of *UbxRNAi* clones. Ubx and GFP immunostains are also shown. GFP+ tissue marks cells expressing *UbxRNAi*. All scale bars shown are 50 micron in size.

expression in the proximal versus distal compartments of the haltere disc. Thus, interestingly, we find that the same CRM is used for both activating and repressive functions within the same tissue.

Previously characterized examples of *Ubx* negative autoregulation [13,16,17] highlight the importance of autoregulation to buffer against increases in Ubx protein level, either from ectopic pulses of transgenic Ubx or from more subtle increases in *Ubx* copy number via chromosomal aberrations containing duplications of the *Ubx* locus [13,16]. In each of these scenarios, an increase in Ubx protein resulted in the complete repression of the endogenous locus or *Ubx* CRMs (as readout by Gal4 insertions) within the locus, respectively.

In contrast to these previous studies, we present a role for negative autoregulation in establishing distinct levels of *Ubx* transcription between compartments within a single tissue during normal development. We molecularly dissect this autoregulatory mechanism using transgenes, where we characterize the contribution of a small cluster of low affinity binding sites that are bound by Ubx and its cofactors Extradenticle (Exd) and Homothorax (Hth) and mediate proximal down-regulation of *Ubx* expression. Further, we show that at the endogenous *Ubx* gene additional binding sites and mechanisms are required to achieve negative autoregulation. Our results underscore the complementary value of transgenic and genome modification approaches to understanding gene regulation, and highlight the increasing importance of interrogating this problem within the complexity of native loci enabled by genome editing technologies such as CRISPR [18].

## Results

### Proximal:Distal (PD) expression bias of Ubx is established at the level of transcription

The importance of *Ubx* expression for haltere fate was established decades ago by the discovery of mutations that resulted in a loss of *Ubx* function in the developing haltere and the concomitant production of four-winged flies [15,19,20]. In these cases, misregulation of the ON/OFF *Ubx* expression switch results in a complete homeotic transformation of the haltere to the wing. Conversely, gain of function of *Ubx* in the developing wing results in a near complete transformation of wing to haltere [8,21]. In addition, more subtle changes in *Ubx* expression level have also been shown to result in malformations of the adult appendage: decreases in *Ubx* dose lead to a slightly enlarged haltere [16] and increases in *Ubx* dose result in decreased adult haltere size–though size is buffered to a certain extent against changes in *Ubx* dose [16,22]. While this establishes the importance of maintaining appropriate levels of *Ubx* expression in a given tissue context, and the presence of autoregulatory responses to genetic perturbation to maintain levels within the appropriate window, it does not address mechanisms present during normal development to establish differential expression levels between compartments in a single tissue.

In third instar larval haltere imaginal discs, *Ubx* is expressed at higher levels in the distal region than in the proximal region (Fig 1A and 1B) offering a system to probe how different expression levels are established. This differential *Ubx* expression pattern can be seen at both the level of total cellular protein (Fig 1B)–quantified to reveal an average distal/proximal ratio of 1.67 (Fig 1B', S1 Fig)–and at the level of nascent RNA production (Fig 1C), providing evidence that the proximal-distal bias is established, at least in part, due to transcriptional control. These findings are additionally corroborated by the presence of a proximal-distal bias in the activity of several Gal4 enhancer-trap lines, including *Ubx-lacZ*^HC166D (Fig 1D, left) [23], which presumably captures input from multiple CRMs required for the establishment of the observed proximal-distal bias. Interestingly, RNAi-mediated clonal reduction of proximal Ubx elevates levels of LacZ driven by *Ubx-lacZ*^HC166D to distal levels (Fig 1D, right). Taken together, these results suggest that the observed proximal-distal bias is established through the active

repression of proximal Ubx levels at the level of transcriptional control and that this occurs, either directly or indirectly, through *Ubx* activity.

## Proximal autorepression of Ubx relies on Hox cofactors Hth/Exd

The establishment of the PD bias through proximally-restricted repression suggests the existence of a factor that is similarly restricted to the proximal compartment. We therefore investigated the potential contribution of the well-established Hox cofactors, Homothorax (Hth) and Extradenticle (Exd). *hth* expression is restricted to the proximal compartment, with the exception of expression in the dorsal, distal hinge (Fig 2A, asterisk), thus largely mirroring the pattern of low *Ubx* (Fig 2A). To test if Hth represses *Ubx* transcription in the proximal haltere, we generated *hth* null (*hth^P2*) clones and performed immunostaining against endogenous Ubx protein. In the absence of Hth, we observed a de-repression of proximal Ubx levels, indicating the involvement of Hth in the downregulation of *Ubx* in the proximal haltere (Fig 2B).

Given the involvement of Hth, we next tested the role of *exd*–which encodes an obligate binding partner of Hth. Hth interacts directly with Exd and is required to translocate Exd from the cytoplasm of the nucleus [24]. In the distal haltere disc, where Hth is absent, Exd is cytoplasmic and seemingly non-functional. To test if Exd works with Hth to repress *Ubx* transcription in the proximal haltere, we generated *exd* null clones with two different alleles (*exd^1* and *exd^2*). As observed for *hth^P2* clones, loss of Exd in the proximal (but not distal) haltere resulted in de-repression of *Ubx* (Fig 2D, S2A Fig). This result contrasts with a previous report showing loss of Ubx protein in *exd^2* mutant clones, suggesting positive regulation of *Ubx* by Exd [25]. To resolve this discrepancy, we further tested the role of Exd by knocking it down in clones expressing *exd* RNAi. In these clones, and in agreement with our *exd* null clones, proximal Ubx levels are elevated (S2B Fig).

Three known isoforms of Hth exist due to alternative splicing of the locus: a full-length isoform that contains a homeodomain and is thus able to bind DNA (Hth^FL) and two homeodomain-less isoforms that cannot directly bind DNA and thus depend on complex formation with other factors for recruitment to DNA (collectively referred to as Hth^HM) [26]. To test if Hth requires its homeodomain for proximal *Ubx* repression, we generated clones of the *hth^100-1* allele, which only produces Hth^HM due to a premature stop codon before the homeodomain [27]. In these clones, and thus in the absence of Hth^FL, Ubx levels are not elevated and the PD bias remains intact (Fig 2C). Thus, although Hth is necessary for proximal *Ubx* repression, direct binding of Hth to DNA is dispensable; HD-less isoforms of Hth are sufficient for the bias in Ubx levels along the PD axis.

Given the evidence for the repressive effects of Hth on *Ubx* expression, we sought to understand if there was mutual repression between the two factors. We generated *Ubx* null (*Ubx^9-22*) mitotic clones and performed immunostaining against Hth protein. In the absence of Ubx, Hth protein levels were unaffected suggesting that, while Hth represses *Ubx*, Ubx does not repress *hth* (Fig 2F). These results were confirmed by immunostaining for Exd protein in *Ubx* null clones (Fig 2E). As Hth is required for the translocation of Exd to the nucleus, the subcellular localization of Exd can serve as a readout for the presence or absence of Hth. In the presence of Hth, Exd is nuclear, but if Hth is lost upon induction of *Ubx* null clones, Exd should become cytosolic. However, contrary to this prediction, the levels of Exd and subcellular localization of Exd are unaffected in *Ubx* null clones (Fig 2E).

## A Ubx-regulated module within the intronic *abx* CRM mediates autorepression

Previous work on the *cis*-regulatory logic of *Ubx* identified two large spatially distinct genomic regulatory regions: a region upstream of the TSS (Upstream Control Region–UCR) that

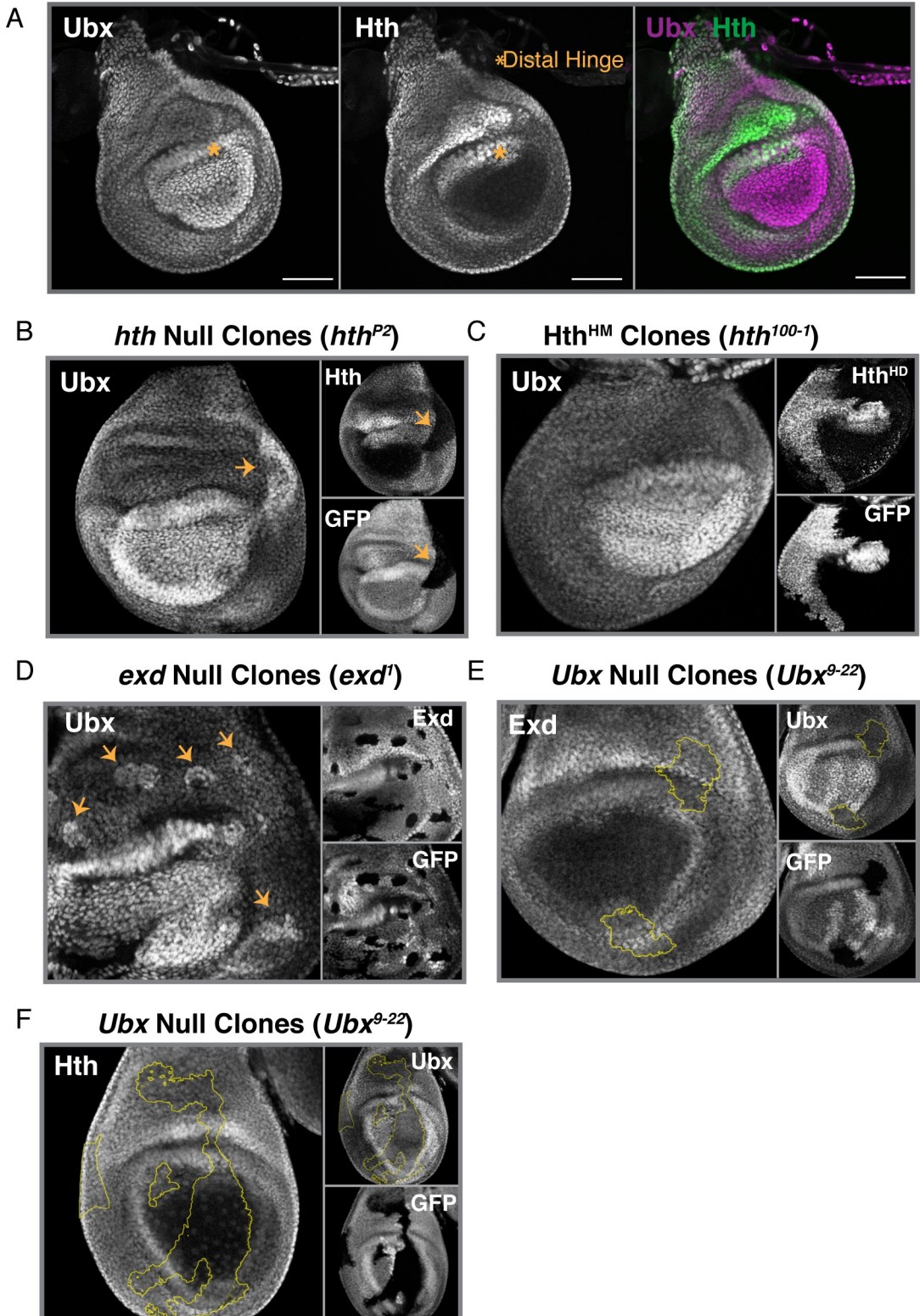

**Fig 2. Proximal autorepression of *Ubx* Relies on Hox cofactors Hth/Exd. (A)** Immunostain of Ubx (Left) and Hth (Middle), along with a merge of the two (Right) in 3<sup>rd</sup> instar haltere imaginal discs. The distal hinge is marked with an asterisk to denote a Ubx-high region that also expresses *hth*. **(B)** Ubx immunostain in haltere discs in which *hth* null clones (*hth^{P2}*) have been induced. Hth and GFP immunostains are also shown. Clones are Hth- and GFP-negative. Yellow arrows point to the clone. **(C)**

Ubx immunostain in haltere discs in which Hth$^{HM}$-only clones (*hth$^{100-1}$*), lacking the C-terminal homeodomain) have been induced. An immunostain for the C-terminus of Hth (Hth$^{HD}$) and GFP are shown. Clones are Hth$^{HD}$- and GFP-negative. **(D)** Ubx Immunostain in haltere discs in which *exd* null clones (*exd$^1$*) have been induced. An Exd and GFP immunostain are also shown. Clones are Exd- and GFP-negative. Yellow arrows point to clones. **(E)** Exd immunostain in haltere discs in which *Ubx* null clones (*Ubx$^{9-22}$*) have been induced. Ubx and GFP immunostains are also shown. Clones are Ubx- and GFP-negative, and outlined in yellow. **(F)** Hth immunostain in haltere discs in which *Ubx*-null clones (*Ubx$^{9-22}$*) have been induced. A Ubx and GFP immunostain are also shown. Clones are Ubx- and GFP-negative, and outlined in yellow. All scale bars shown are 50 micron in size.

regulates *Ubx* in parasegment 6 (PS6) and the posterior haltere, and a region within the large third intron of *Ubx* (Downstream Control Region–DCR) that regulates *Ubx* in PS5 and anterior haltere [11,14]. While a 35 kb UCR fragment (35UZ) drives reporter expression in the posterior haltere, various fragments derived from the DCR, termed the *anterobithorax* (*abx*) CRM, drive reporter expression throughout the haltere that recapitulates *Ubx* expression, including the presence of a PD bias [11,13]. In addition, classically defined alleles of the *Ubx* locus affecting anterior expression in the haltere are located within the *abx* region [7,12]. Smaller fragments of *abx* of 6.8 kb (*abx6.8*) and 3.2 kb (*abx3.2*) were sufficient to recapitulate the *Ubx* expression pattern, including the PD bias, in the haltere (Fig 3A) [11]. These fragments also drive reporter expression in other imaginal discs where *Ubx* expression is absent, presumably due to the absence of a necessary repressive element, such as a Polycomb Response Element (PRE). Notably, the presence of a PD bias in the wing imaginal disc suggests that proximal repression of these CRMs is not Ubx-specific and can also be carried out by the Hox protein, Antennapedia (Antp, Fig 3C), which, together with *hth* and *exd*, is expressed in the proximal compartment of the wing disc.

That the *abx* region functions as a *cis*-regulatory module is further corroborated by published FAIRE-seq data [28] that shows a broad region of open chromatin particularly within the domains of the two small *abx* fragments, *abx6.8* and *abx3.2* –hereafter called *abxFAIRE* (Fig 3A). Not only is this region specifically accessible in the haltere as compared to the wing (Fig 3A, track 1 versus track 2), it is also bound by Ubx and Hth, as assayed by previous ChIP-chip experiments in haltere imaginal discs (Fig 3A, track 3 and 4) [28,29]. Consistent with these earlier results, clonal deletion of *abxFAIRE* results in a decrease in Ubx levels–ranging from a slight reduction to complete loss–in both proximal and distal cells throughout the anterior of the haltere disc (Fig 3B, S3B and S3B' Fig) [9]. Despite this activating activity, *abx*-driven reporter expression maintains the PD bias, suggesting that this element also contains sequences required for down-regulation of *Ubx* in the proximal haltere.

Within *abxFAIRE* we focused on a ~1.4 kb fragment (*abxF*) that included the highest enrichment for both Ubx and Hth binding in ChIP experiments (Fig 3A). As expected based on prior reports, *abxF-lacZ* reporter constructs recapitulate *Ubx* expression throughout the haltere, including the PD bias (Fig 3C) [11]. Similar to native *Ubx* levels, reporter expression driven by *abxF* is elevated in the absence of either Hth or Exd, as determined by examining mitotic clones null for each gene (Fig 3D and 3E). Further, as with native *Ubx* levels, proximal repression of *abxF* activity does not require the full-length isoform of Hth (Fig 3F), confirming that direct DNA binding of Hth is unnecessary for the PD bias. Finally, *Ubx* null mitotic clones (*Ubx$^{9-22}$*) similarly results in a de-repression of proximal *abxF* activity to levels comparable to that in the distal pouch (Fig 3G). Taken together, these results confirm that the *abxF* fragment is able to capture the input of Ubx-Hth-Exd in proximal repression and the concomitant establishment of the PD bias, providing evidence for a dual role for *abx* in mediating both activation (distally and proximally) and partial repression (proximally). Thus, *abxF* provides a platform to more finely dissect the mechanism of partial auto-repression by Ubx and its cofactors.

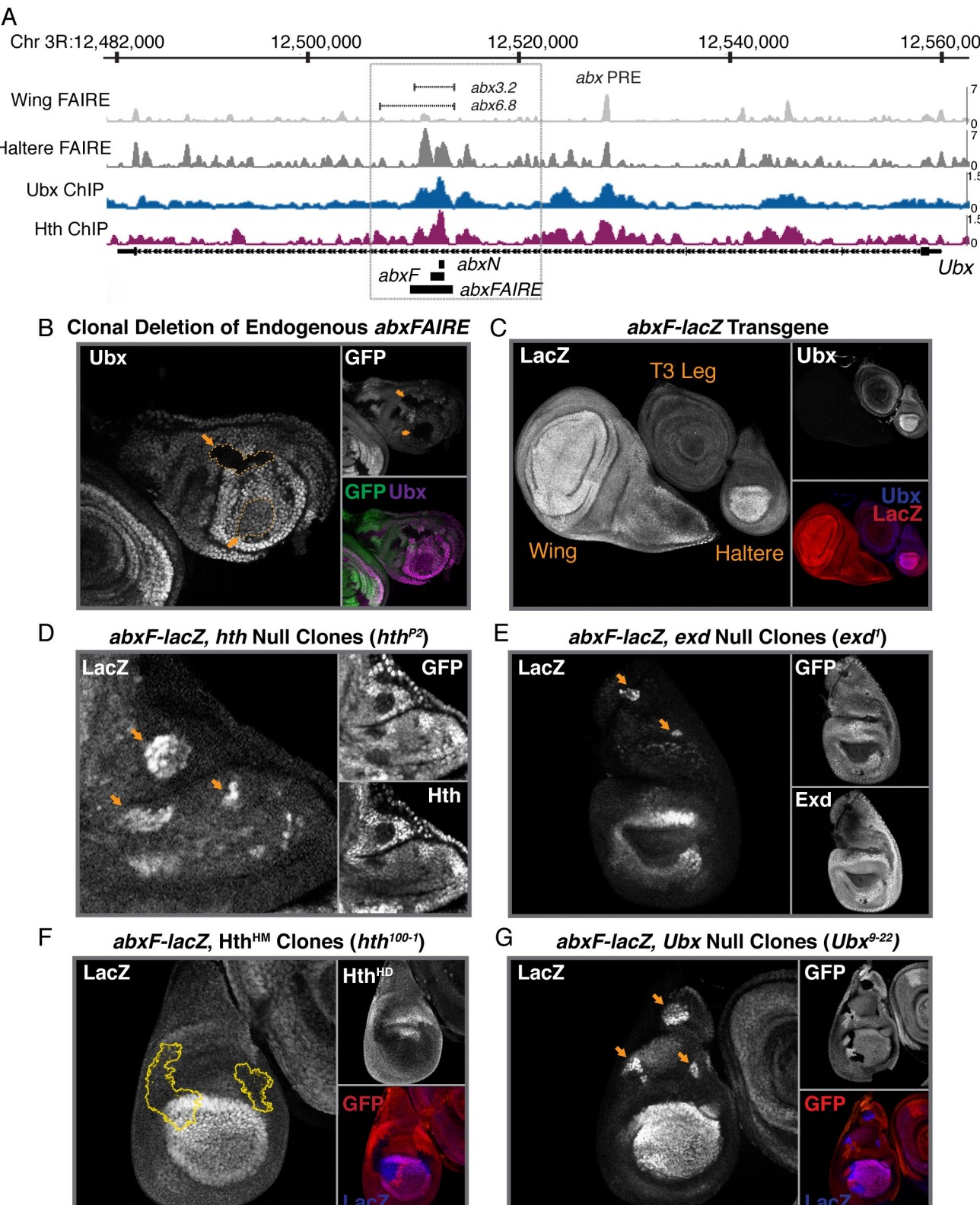

**Fig 3. A Ubx-regulated module within the intronic *abx* CRM mediates autorepression.** (A) A screenshot of the *Ubx* locus within the Integrative Genomics Viewer (IGV) is shown. FAIRE accessibility peaks from [28] are shown for 3rd instar wing and haltere imaginal discs (top two tracks). Ubx and Hth ChIP-chip in 3rd instar haltere discs from [29] are shown (bottom two tracks). The intronic *anterobithorax* (*abx*) regulatory region we focused on is

boxed. The location of two *abx* fragments that drive reporter expression (*abx3.2, abx6.8*) are shown, along with the annotated location of the *abx* Polycomb repressive element (PRE). Regions of interest (*abxN, abxF, abxFAIRE*) in this paper are shown beneath the annotated *Ubx* gene. (B) Ubx Immunostain in haltere discs containing ΔabxFAIRE clones. A GFP immunostain, in addition to a merge of GFP/Ubx, is also shown. Clones are marked by yellow arrows and outlined in yellow. (C) LacZ immunostain in 3rd instar imaginal discs expressing *lacZ* from an *abxF-lacZ* transgene. A wing, T3 leg and haltere are shown. A Ubx immunostain, in addition to a merge of LacZ/Ubx, is also shown. (D) LacZ immunostain in *abxF-lacZ* transgenic haltere discs in which *hth* null clones (*hth^{P2}*) have been induced. GFP and Hth immunostains are also shown. Clones are GFP- and Hth-negative, and marked with a yellow arrow. (E) LacZ immunostain in *abxF-lacZ* transgenic haltere discs in which *exd* null clones (*exd^1*) have been induced. GFP and Exd immunostains are also shown. Clones are GFP- and Exd-negative, and marked with a yellow arrow. (F) LacZ immunostain in *abxF-lacZ* transgenic haltere discs in which Hth^{HM}-only clones (*hth^{100-1}*, lacking the C-terminal homeodomain) have been induced. An immunostain for the C-terminus of Hth (Hth^{HD}) and GFP are also shown. Clones are Hth^{HD}- and GFP-negative, and outlined in yellow. (G) LacZ immunostain in *abxF-lacZ* transgenic haltere discs in which *Ubx* null clones (*Ubx^{9-22}*) have been induced. A GFP immunostain, in addition to a merge of GFP/LacZ, is shown. Clones are GFP-negative and marked with a yellow arrow.

## Ubx directly binds *abx* to auto-repress in the proximal compartment

Given that the *abxF* CRM recapitulates *Ubx* expression bias along the PD axis, and that ChIP-chip results indicate that Ubx is bound at this location, we attempted to identify a Ubx binding site in *abxF* [29]. As a first step, we minimized the 1.4 kb *abxF* fragment and identified a 531 bp minimal autoregulatory CRM, *abxN*, which recapitulates the expression pattern of the larger *abxF* fragment, and *Ubx* itself (Fig 4A) when driving a *lacZ* reporter. Using the recently developed versatile maximum likelihood framework, *No Read Left Behind* (*NRLB*), to predict Ubx-Exd-Hth^{HM} binding, we identified two clusters of low affinity Ubx-Exd-Hth^{HM} binding sites (with predicted maximum relative affinities of 1.28E-3 and 6.94E-3, respectively, Fig 4B, 4B' and 4B'') [30]. Mutation of each binding site cluster to abrogate Ubx binding (resulting in a > 450-fold decrease in relative affinity, Fig 4B' and 4B'') followed by immunostaining of the *lacZ* reporter revealed a decrease in PD bias, though this was more severe following mutation of Cluster 1 (Fig 4C, quantified in 4C'). In fact, mutation of both clusters together exhibited a PD bias defect comparable to that of Cluster 1 alone (Fig 4C, quantified in 4C'). Using both *in vitro* and *in vivo* methods, we next established that Ubx binds directly to *abxN* together with its cofactors. *In vitro* EMSAs revealed the binding of Ubx-Exd-Hth^{HM} to a probe containing the Cluster 1 binding sites. Importantly, this binding was lost upon mutation of these sites (S4B Fig). *In vivo*, ChIP-qPCR was used to demonstrate Ubx binding to both the endogenous *abxN* CRM (consistent with the ChIP-chip data) and the transgenic CRM-*lacZ* reporter gene [29]. Mutation of Cluster 1 binding sites resulted in decreased occupancy of Ubx at transgenic *abxN* (S4C Fig). Further, *Ubx* null clones generated in the background of a Cluster 1-mutated *abxN-lacZ* transgene revealed no further de-repression (S4A Fig), providing additional evidence that the repressive activity of Ubx occurs through direct binding of *abxN* via Cluster 1 binding sites.

Notably, although these data demonstrate that the reduction in PD bias results in part from an increase in proximal expression, mutation of the Ubx-Exd-Hth^{HM} binding sites in *abxN-lacZ* also decreases distal reporter expression levels, particularly within the distal hinge region (Fig 4C, S4D and S4D' Fig). These results suggest the existence of positive inputs in the distal compartment that are compromised by these mutations. However, a similar effect on distal levels is not observed when Ubx binding sites are mutated in a larger *abx* fragment (*abxF*), although proximal derepression is observed in this context (see below). Additional studies will be needed to determine if the observed decrease in distal expression of *abxN* is mediated by Ubx and if it is relevant to native *Ubx* regulation.

All of the predicted binding sites within *abxN*, including those found within Cluster 1, are low affinity binding sites, defined as approximately 0.1% of the maximal binding site affinity in the genome [31]. Interestingly, when we mutate Cluster 1 to contain a high affinity binding site as determined by *in vitro* SELEX-seq experiments and *NRLB* predictions (a 200-fold

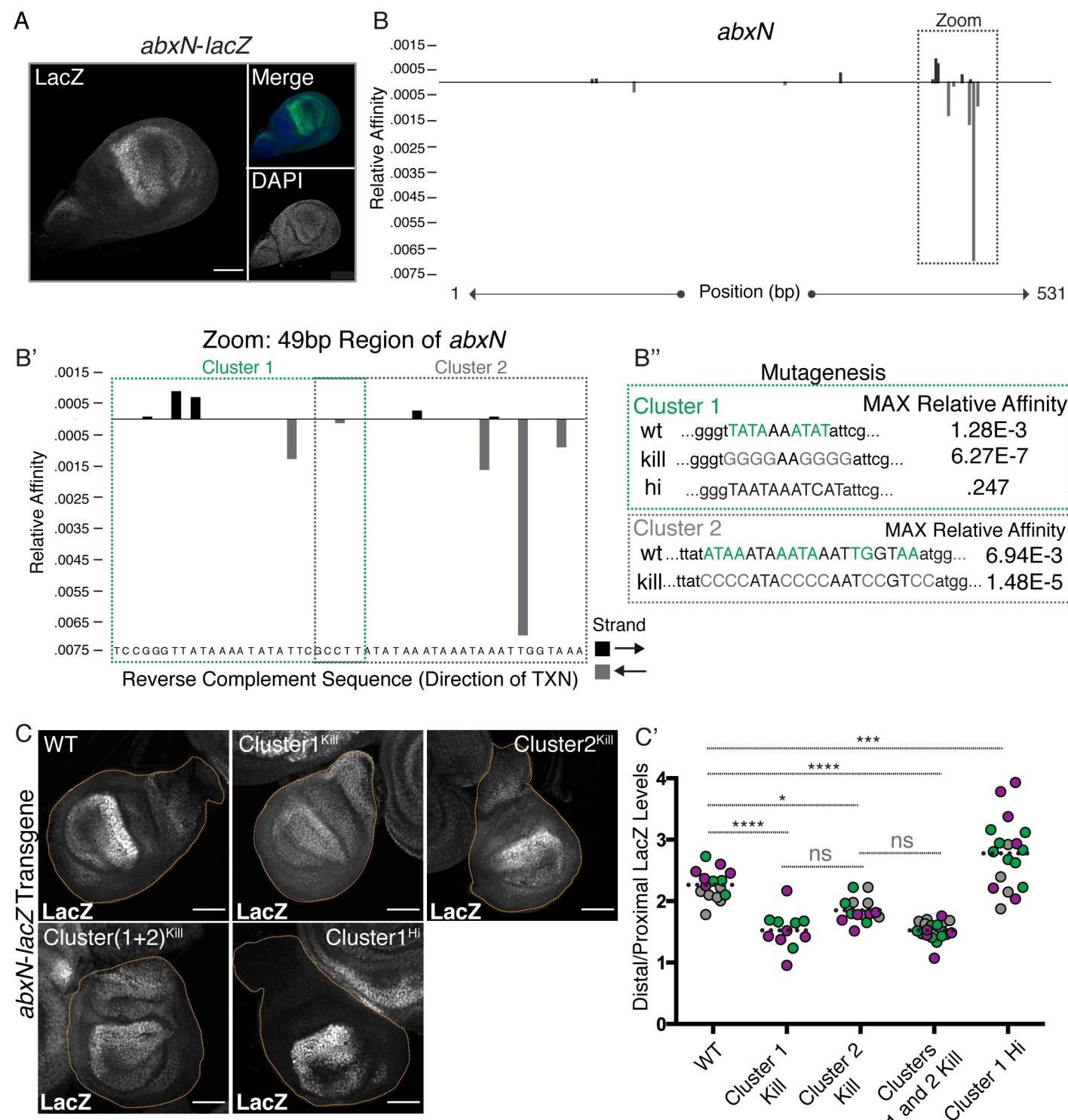

**Fig 4. Ubx directly binds *abxN* to autorepress in the proximal compartment.** (A) LacZ immunostain in *abxN-lacZ* haltere discs. A DAPI nuclear stain and a merge of LacZ/DAPI are shown. (B) *NRLB* predicted Ubx-Exd binding sites within the 531 bp *abxN*, along with their relative affinities. Strandedness of the predicted binding sites is depicted by color: black bars above the axis denote the start of a site in the forward direction and gray bars below the axis denote start of a site in the reverse direction. The boxed region is focused on in B'. (B') 49 bp zoomed in region from B. Binding sites are divided into Cluster 1 and Cluster 2. (B") Schematic of mutations of Cluster 1 and 2 made in this paper. Cluster 1 was mutated to abrogate Ubx/Exd binding (Kill) and enhancer Ubx/Exd binding (Hi). Cluster 2 was mutated to abrogate Ubx/Exd binding (Kill). (C) LacZ immunostains in *abxN-lacZ* transgenic haltere discs. Exemplary images are shown for discs transgenic for wildtype and mutant forms of the transgenic *abxN*. (C') Quantification of the Distal:Proximal ratio in discs described in C. Colors indicate discs from a single

experiment. Dotted bar signifies the mean value. Significance values are derived from one-way ANOVA analysis followed by Tukey's multiple comparisons test with alpha = .05. Multiplicity adjusted p-values for each level of significance are: < .0001 for ****, .0005 for ***, and .0143 for *. All scale bars shown are 50 micron in size.

increase in affinity to .247; Fig 4B"), levels of the *lacZ* reporter within the proximal compartment are decreased and the PD bias is increased accordingly (Fig 4C and 4C') [31]. Not only does this result support the autoregulatory role of Ubx in controlling the PD bias–increasing the binding affinity results in greater proximal repression–but it also suggests the functional importance of low affinity binding sites [32,33]. In this case, low affinity binding sites may allow for a more tunable state of transcription that is sensitive to changes in TF levels; Ubx-Exd-Hth[HM] binding sites in *abx* appear to be optimized to produce the correct amount of *Ubx* repression.

## Mutation of *abxN* at the native locus is not sufficient to alter the proximal-distal bias

Having established the proximally-restricted repressive role of Ubx on *abxN* to form the PD bias in a transgenic context, we sought to understand this phenomenon at the endogenous *Ubx* locus. To this end we devised a two-step strategy that utilizes CRISPR/Cas9 genome engineering along with PhiC31 based recombinase mediated cassette exchange (PhiC31 RMCE) to replace the native, wildtype *abx* CRM with versions containing Ubx-binding site mutations (Fig 5B). Two replacements were performed: one that encompasses the entirety of *abxFAIRE*, including *abxF* and *abxN* (*Targeted Region*[4kb], ~4 kb), and a second that encompasses *abxN* (*Targeted Region*[2kb], ~2.2 kb, Fig 5A). These "*abx*-replacement platforms," verified by both PCR and Southern Blot analysis, were then used to re-insert *abx* sequences (via RMCE) with and without the desired Ubx-Exd binding site mutants (Fig 5B). Because this is not a scarless editing strategy, it was important to confirm that the sequences remaining as a product of attP/attB recombination did not affect *Ubx* expression. To test this, we used RMCE to reinsert the wildtype sequence of each targeted region (*Targeted Region*[2kb] in Fig 5 and S5 Fig; *Targeted Region*[4kb] in Fig 6 and S6 Fig) such that the only difference between the engineered allele and the native allele is the presence of the recombinase sites. Both wildtype *abx* replacement alleles were able to homozygose, and homozygous flies developed normally with no noticeable defect. In addition, the generation of clones that are homozygous for the wildtype *abx* replacement alleles showed no differences in *Ubx* expression compared to neighboring wild type cells, as assayed at the protein level by immunostaining (S5A Fig, top panel and S6B Fig, top panel).

Having established that these RMCE platforms do not affect *Ubx* expression, we next sought to understand the effect of Ubx binding site mutants within the context of the *Ubx* locus on proximal repression and the formation of the PD bias. Because the mutation of Cluster 1 binding sites showed a greater reduction in the PD bias in our transgenic studies, we first used our 2 kb replacement platform (Targeted Region[2kb]) to generate *abxN* replacement alleles containing mutations to either abrogate Ubx-Exd binding (*abxN-Cluster1*[Kill]) or enhance Ubx-Exd binding (*abxN-Cluster1*[Hi]). While each of these mutations altered the PD bias in the context of the transgenic *lacZ* reporter, neither resulted in a detectable change in the PD bias of Ubx, as assayed at the protein level. This was true whether we assayed *Ubx* expression in haltere discs homozygous for the mutant alleles (Fig 5C, quantified in 5C') or through the generation of homozygous mutant clones (S5A Fig). Thus, mutation of a single cluster of low affinity Ubx-Exd-Hth[HM] binding sites within the >100 kb *Ubx* locus is insufficient to impact Ubx expression.

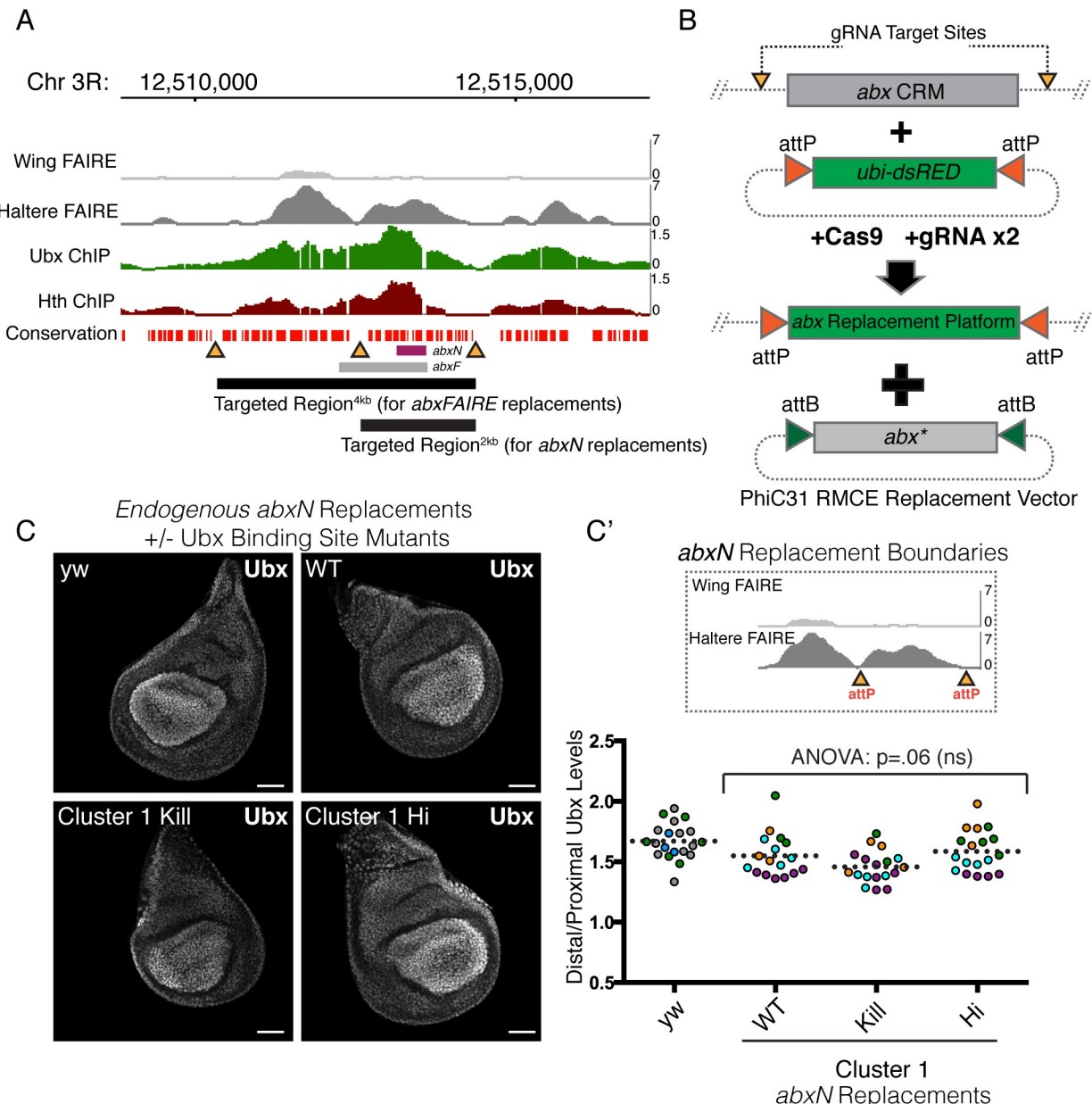

**Fig 5. Mutation of *abxN* at the native locus is not sufficient to alter the proximal-distal bias.** **(A)** (Top) IGV Genome Browser Snapshot displaying FAIRE accessibility peaks in the wing and haltere [28], and Ubx and Hth ChIP-chip peaks in the haltere [29]. The bottom conservation track was downloaded from UCSC Genome Browser and denotes the evolutionary conservation of *D. melanogaster* sequence with twelve other *Drosophila* species, mosquito, honeybee and red flour beetle. Regions devoid of bars lack conservation. (Bottom) Regions targeted with CRISPR/Cas9. Black bars are the regions that were targeted for deletion and/or replacement. Yellow triangles denote regions of low conservation that were targeted by the Cas9/gRNA complex. These are also the points of insertion of the integrase recognition sequence (*attP*). **(B)** A schematic of the two-step CRM replacement strategy. gRNA sites were chosen in non-conserved regions surrounding the *abx* sequence of interest. A dual-gRNA expressing plasmid was injected along with a donor cassette containing an attP-flanked fluorescent selection marker (*ubi-dsRED*) into *nanos-Cas9* flies. The resulting "*abx* Replacement Platform" serves as a means to delete or insert modified versions of *abx* using PhiC31-based RMCE. **(C)** Ubx immunostain in haltere discs homozygous for *abxN* replacement alleles. WT and Cluster 1 mutants are shown alongside a yw control. **(C')** (Top) A schematic of the Targeted Region$^{2kb}$ replacement boundaries for discs shown in **C**. (Bottom) Quantification of the Distal:Proximal ratio in discs from **C**. Each point is an individual disc. Points are color-coded by experiment. Dotted line signifies mean value. Significance was tested using a one-way ANOVA analysis followed by Tukey's multiple comparisons test with alpha = .05. All scale bars shown are 50 micron in size.

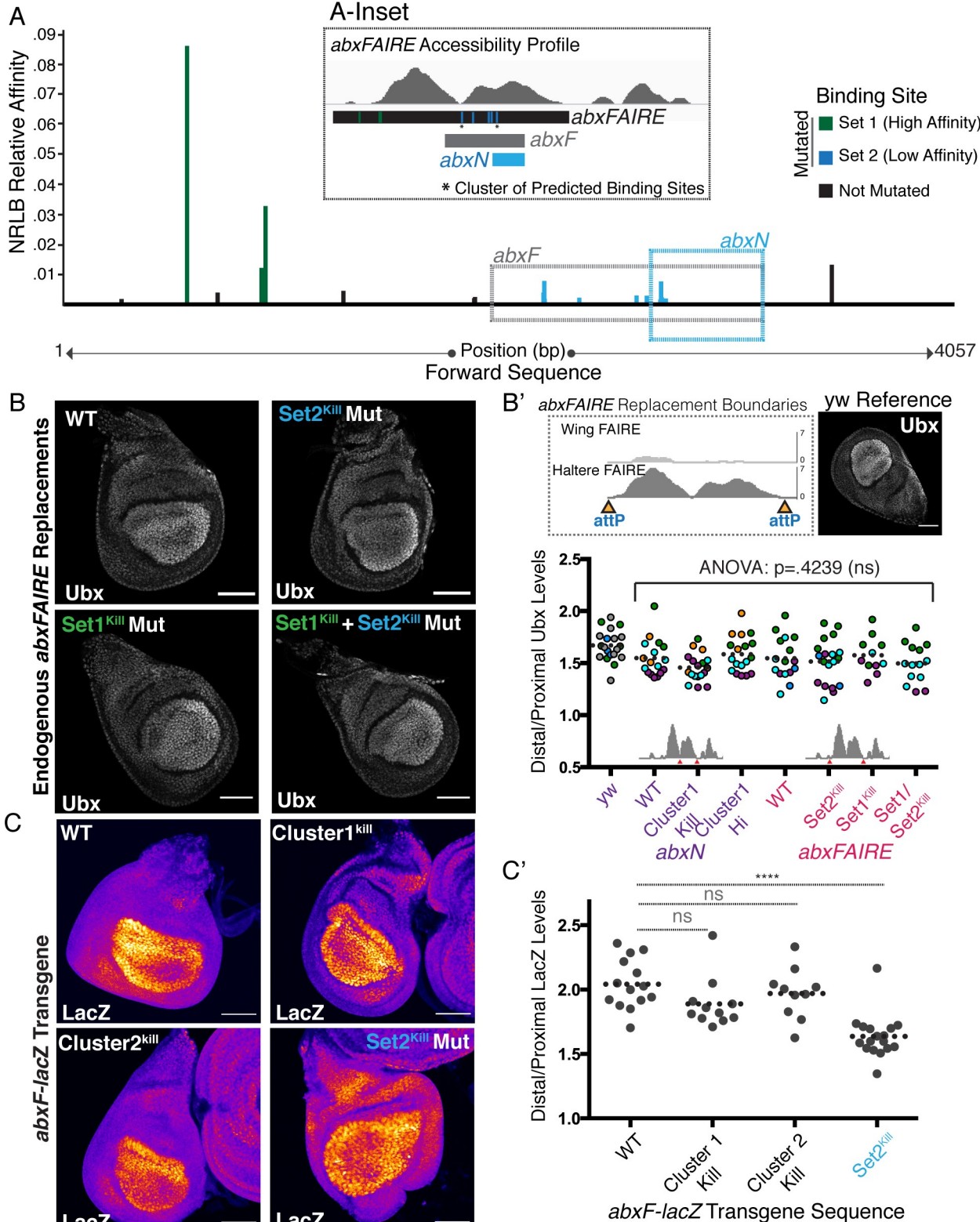

**Fig 6. Multiple low affinity Ubx/Exd binding sites are required for PD bias formation. (A)** The top twenty *NRLB* predicted Ubx-Exd binding sites within the 4057 bp *abxFAIRE*, along with their relative affinities. Sites are segregated into high affinity (Set1, green), low affinity (Set2, blue), both of which are mutated. Sites in black were not mutated. Locations of *abxN* and *abxF* are boxed. The *inset* places the mutated sites in the context of haltere *abx* accessibility. **(B)** Ubx immunostain in haltere discs homozygous for *abxFAIRE* replacement alleles. WT, Set2^Kill Mutants, Set1^Kill

Mutants, and the combination are shown. **(B')** (Top) A schematic of the Targeted Region[4kb] replacement boundaries for discs shown in **B**, along with a Ubx immunostain in haltere discs from *yw* flies for control. (Bottom) Quantification of the Distal:Proximal (D/P) ratio in discs from **B** combined with *abxN* replacements from Fig 5. Each point is an individual disc. Points are color-coded by experiment. Dotted line signifies mean value for each genotype. Significant differences between each of the replacement alleles was tested using one-way ANOVA analysis followed by Tukey's multiple comparisons test with alpha = .05. **(C)** LacZ immunostain in haltere discs with a wildtype or mutant *abxF-lacZ* reporter transgene (using the FIRE LUT in FIJI). **(C')** Quantification of Distal:Proximal ratios in *abxF-lacZ* transgenic discs. Each point denotes an individual disc. Dotted line denotes mean value for each genotype. Significance values are derived from one-way ANOVA analysis followed by Tukey's multiple comparisons test with alpha = .05. Multiplicity adjusted p-values for each level of significance are: $< .0001$ for ****. All scale bars shown are 50 micron in size.

## Multiple low affinity Ubx-Exd binding sites are required for PD bias formation

The disparity between the effect of Cluster 1 mutations on the PD bias of the CRM *lacZ* reporter transgene and the absence of an effect of the same mutations at the endogenous locus suggested the existence of additional regulatory inputs; as mentioned above, the *Ubx* locus itself is >100 kb, *abxFAIRE* is ~4 kb, and *abxF* is ~1.4 kb, whereas *abxN* contains only 531 bp of *abx* sequence. Thus, the difference could suggest the involvement of multiple Ubx-Exd binding sites in Ubx-mediated proximal repression that are present in the greater *abx* region, but not within the *abxN* transgene. In fact, we can see some evidence of this even within the *abxN* transgenic CRM, itself. Even though the abrogation of binding to Cluster 1 resulted in the greatest amount of proximal de-repression, mutation of Cluster 2 also showed some effect (Fig 4C and 4C'). Thus, we sought to expand our window outside of *abxN* to search for additional predicted Ubx-Exd binding sites. Using *NRLB* [30], we identified the top twenty predicted Ubx-Exd binding sites within *abxFAIRE*, ranked by relative affinity (Fig 6A). Very few high affinity sites are present (Set1 (High Affinity), Fig 6A), and these binding sites exist near the border of the FAIRE peak, suggesting that they are not highly accessible *in vivo* (Fig 6A-Inset); the remaining predicted Ubx-Exd binding sites are low affinity sites (Set2 (Low Affinity), Fig 6A), on par with those found within the smaller *abxN* fragment.

To streamline our mutagenesis efforts, we focused our attention on 1) mutating the small set of high affinity binding sites (Set1, Fig 6A), and 2) a larger set of low affinity binding sites all residing within *abxF* (Set2, Fig 6A). Mutations were made to abrogate Ubx-Exd-Hth[HM] binding within each of these sets independently and together, resulting in three mutant genotypes (S6A Fig). In order to generate *abxFAIRE* mutants at the endogenous locus we utilized our 4 kb replacement platform (Targeted Region[4kb], Fig 5A and 5B) to perform RMCE. As above, we confirmed with an *abxFAIRE*[WT] replacement that the presence of the RMCE scars does not affect *Ubx* expression (S6B Fig, top panel).

A comparison of the PD bias in flies homozygous for the *abxFAIRE* replacement alleles revealed no defect in proximal repression upon mutation of either the set of high (Set1[Kill]), low (Set2[Kill]), or combined high and low binding sites (Set1[Kill] + Set2[Kill]; Fig 6B, quantification shown along with *abxN* replacements in 6B'). Thus, just as with the *abxN*-Cluster 1 mutants, mutation of up to fourteen binding sites (S6A Fig) within *abxFAIRE* did not detectably perturb the PD bias. This result was confirmed by generating homozygous *abxFAIRE* mutant mitotic clones in developing halteres; again, levels of proximal *Ubx* expression remained comparable to wildtype (S6B Fig).

Because not all low affinity binding sites within *abxFAIRE* were mutated–including those with relative affinities within the top 20 selected sites (Fig 6A, shown in black) and those below the relative affinity cutoff used for NRLB (not represented in figure)–it remained possible that the lack of effect observed upon mutation of endogenous *abxFAIRE* was due to the presence of additional, unmutated binding sites. To address this, we first sought to understand the effect of

additional *abx* mutations in a transgenic CRM reporter gene. We focused our attention on the set of low affinity *abxF* binding sites (Set2, Fig 6A) and returned to our *abxF-lacZ* reporter transgene to conduct an analysis of PD bias as a result of increasing mutation load on the ~1.4 kb *abxF* CRM. While mutation of the Cluster 1 or Cluster 2 binding sites alone did not result in proximal de-repression in this context, abrogation of binding to the set of all low affinity binding sites (Set2$^{Kill}$, Fig 6A) resulted in a slight, but statistically significant, de-repression and a concomitant reduction in the PD bias (Fig 6C, quantified in 6C'). The requirement for many mutations to begin to observe de-repression in the context of a longer *abx* fragment supports the idea that multiple low affinity binding sites are necessary to achieve *Ubx* negative autoregulation. This also helps resolve the disparity between our results with the transgenic *abx-lacZ* reporters and at the endogenous *Ubx* locus. The presence of even more binding sites outside of the *abxFAIRE* region could also contribute to PD bias formation, masking the de-repression effect of mutations made on only a subset of binding sites.

To further assess the existence of binding sites outside of *abxN*, we replaced the endogenous ~4 kb *abxFAIRE* region with the much shorter (531 bp) *abxN* with and without Cluster 1 binding sites mutated, effectively deleting ~3.5 kb of *abxFAIRE* (S7A and S7B Fig). A reduction of the total number of potential Ubx-Exd binding sites within the *abxFAIRE* region through this strategy holds the potential to distinguish between two conclusions: (1) if an impact of mutating Cluster 1 on the PD bias is observed, it would argue for the sufficiency of binding sites within *abxFAIRE* for PD bias formation, or (2) if there is no impact of mutating Cluster 1, it is likely that additional Ubx-Exd-Hth$^{HM}$ binding sites outside of the *abxFAIRE* region are involved in PD bias formation. Two phenotypes are observable upon replacement of *abxFAIRE* with the shorter *abxN*. First, for both the WT *abxN* and Cluster 1$^{kill}$ replacements, large patches of tissue exhibit no *Ubx* expression (S7B Fig). Thus, minimization of the *abx* CRM results in a seemingly stochastic loss of *Ubx* activation, underscoring the activating role of *abx*. Second, while this phenotype makes it more challenging to address the effects of the *abxFAIRE-abxN* replacements on PD bias, in cells where *Ubx* expression was present, the PD bias remained intact even upon mutation of *abxN* Cluster 1 (S7B Fig). This result confirms the likely involvement of binding sites elsewhere in the locus (outside of *abxFAIRE*) in *Ubx* negative autoregulation.

## Depletion of Ubx protein de-represses proximal Ubx transcription at the native locus

While we were able to establish the requirement for the cofactors, Hth and Exd, for proximal repression of the native *Ubx* locus through the generation of mutant mitotic clones (Fig 2B and 2D), evidence for the involvement of Ubx protein, itself, was restricted to the use of *abx-lacZ* transgenes (Fig 3 and Fig 4) and the *lacZ*$^{HC166D}$ enhancer trap (Fig 1). To address the role of Ubx protein at the native *Ubx* locus, we made use of the nanobody-based deGradFP system designed to direct depletion of a GFP fusion protein using a genomically encoded and spatially restricted anti-GFP nanobody coupled to a ubiquitin E3 ligase that leads to proteasomal degradation of the targeted fusion protein [34,35]. In parallel, we generated two *Ubx* knockin alleles through a two-step CRISPR/RMCE targeting strategy that replaced *Ubx exon 1* with either a *GFPUbx exon 1* protein fusion (GFPUbx) or a *GFP-t2a-Ubx Exon 1* allele (*GFP-t2a-Ubx*) that produces both GFP and Ubx protein in a 1:1 ratio (S8 Fig). By combining the deGradFP system with these GFPUbx alleles, along with smRNA FISH to assay Ubx transcription, we could directly interrogate whether a reduction of Ubx protein results in proximal de-repression of the endogenous *Ubx* locus.

Restriction of the expression of the deGradFP nanobody (Nslmb-vhhGFP), and thus depletion of GFPUbx, was achieved both spatially using the Gal4/UAS system, and temporally by addition of a temperature-sensitive Gal80 repressor. For our purposes, degradation of Ubx was restricted to either the distal compartment (using *nubbin-Gal4*) or to the proximal compartment (using *teashirt-Gal4*), and allowed to occur for 24 hours prior to dissection through a temperature shift from 18˚C to 30˚C (Fig 7A). smRNA FISH of homozygous *GFPUbx* haltere discs reveals the expected PD bias with proximal transcription levels lower than distal. This can also be observed by assessing native GFP fluorescence from the fusion gene (Fig 7B, top). Degradation of GFPUbx induced distally did not affect *Ubx* transcription as assayed by smRNA FISH (Fig 7B, middle left), further corroborated by GFP intensity in these discs. Distal GFPUbx protein levels decreased almost to proximal levels because, though transcription remains the same, the resulting protein is continually degraded by the nanobody (Fig 7B middle right). In contrast, degradation of GFPUbx proximally resulted in increased proximal transcription as assayed by smRNA FISH (Fig 7B, left). Thus, consistent with our findings using transgenic *abx* reporters, Ubx is necessary for proximal repression at the endogenous locus; depletion of Ubx protein results in *Ubx* de-repression. This negative autoregulatory loop, though, appears restricted to the proximal compartment as loss of Ubx distally does not result in increased distal transcription levels. Finally, to ensure that these effects were due to the loss of Ubx, we repeated the same experiments on the *GFP-t2a-Ubx* allele. Here, degradation of GFP occurs independent of Ubx (Fig 7A); and, although clear loss of GFP intensity can be observed due to expression of the deGradFP nanobody, no effect on transcription from the *GFP-t2a-Ubx* allele was seen either distally or proximally (Fig 7C).

## Discussion

The proper regulation of gene expression levels is critical for proper cell function and development, and is particularly true for developmental genes that must be tightly regulated both in space and time. In this study, we have combined transgenic assays with genome engineering of a native CRM to characterize the mechanism of quantitative transcriptional tuning of a single Hox gene, *Ubx*. Using transgenic CRM reporter assays we have revealed a novel negative autoregulatory mechanism to partially repress proximal Ubx levels relative to distal, which is mediated, at least in part, through the intronic CRM, *abx*. Ubx in complex with its cofactors, Hth and Exd, binds a cluster of low affinity binding sites within *abx* to dampen, but not eliminate, transcription. In truncated *abx* transgenes, the number of binding sites that need to be mutated in order to observe de-repression increases as the length of the CRM–and thus the number of potential low affinity binding sites–increases. However, in our most truncated version (*abxN)*, in which only two clusters of binding sites are predicted, mutation of a single cluster (Cluster 1) that abrogates Ubx binding is sufficient to result in de-repression; and, conversely, mutation of the same cluster to a high affinity binding site enhances proximal repression–thus establishing a connection between the affinity of the Ubx-Exd–Hth[HM] binding site and the strength of repression. While previous reports demonstrated the utility of clusters of low affinity binding sites to balance specificity and robustness [32,36], which is also likely at play here, our data suggest that low affinity binding sites may also be a means to tune transcription levels, thus allowing CRMs to be more responsive to differences in TF concentration [37]. In this case we suggest that the affinity of the Ubx-Exd-Hth[HM] binding sites is tuned to create the right amount, and not too much, repression.

Although it has not yet been feasible to establish a phenotypic consequence for *Ubx* down-regulation in the proximal haltere disc, it is noteworthy that a similar PD bias of Hox expression levels appears to exist in developing butterfly wings [38], suggesting that this

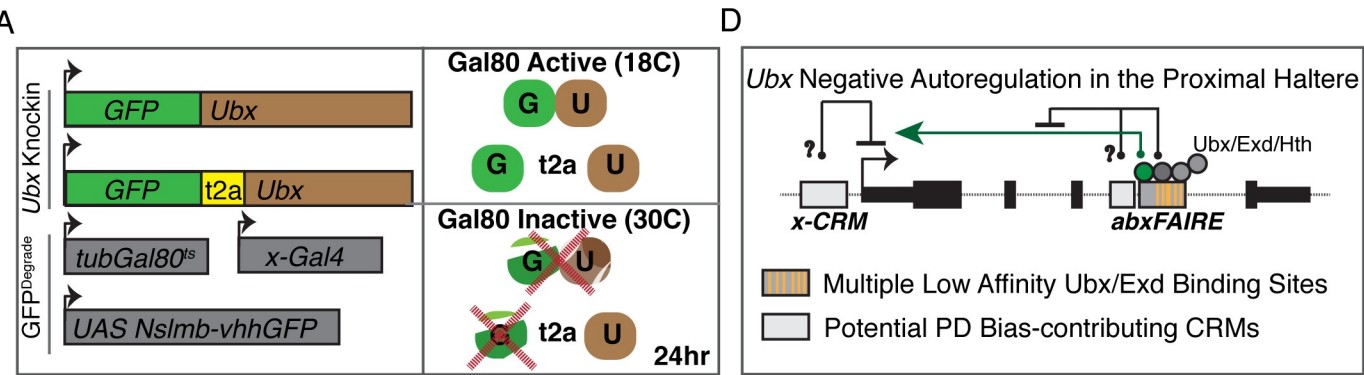

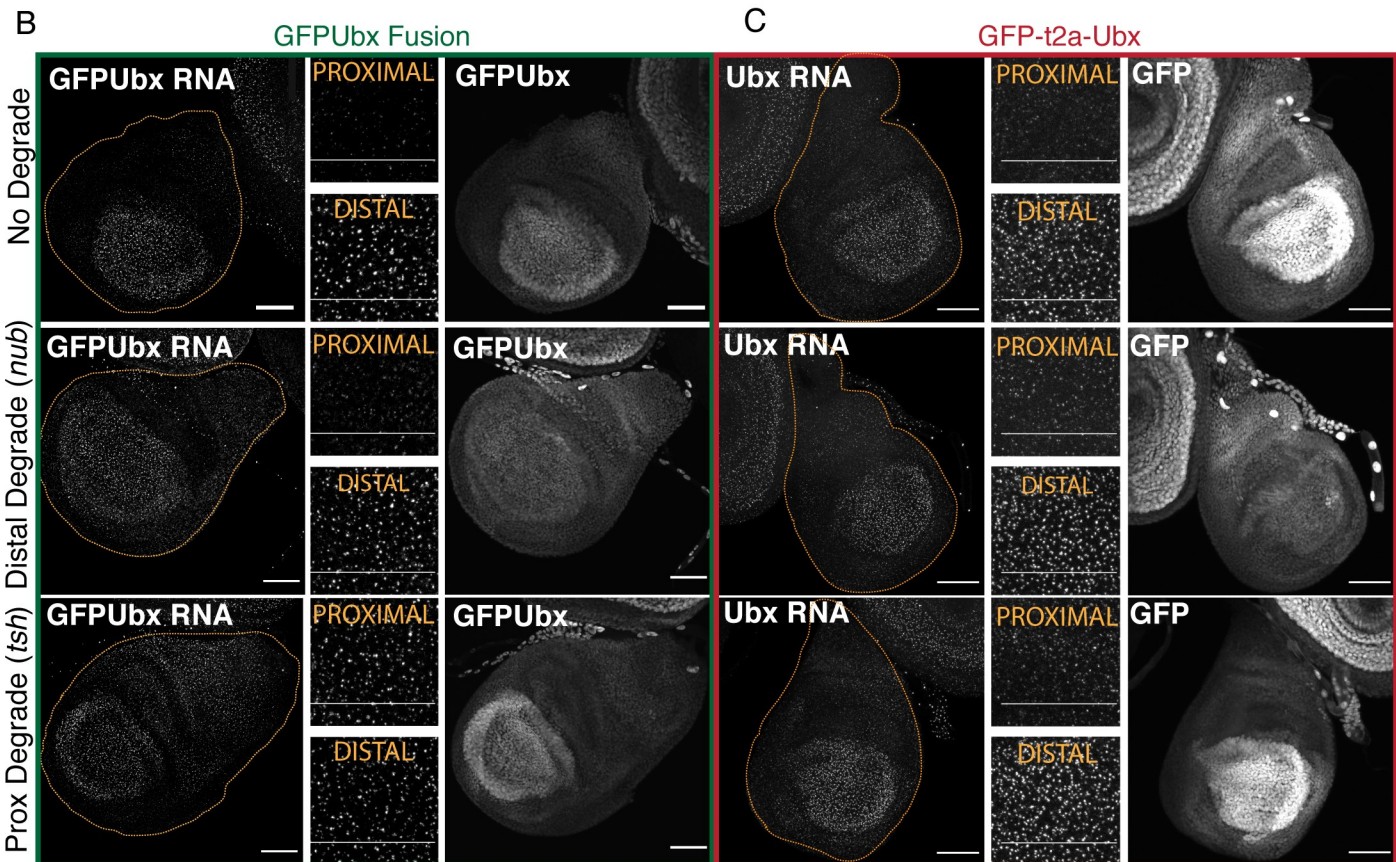

**Fig 7. Depletion of Ubx protein derepresses proximal *Ubx* transcription at the native locus.** (A) Schematic of the GFP-degrade system used. A *GFPUbx* fusion and a *GFP-t2a-Ubx* knockin were generated by CRISPR. These *Ubx* alleles were coupled with a UAS-driven nanobody-based GFP degrade system (from [34,35]), compartment restricted Gal4 drivers, and a temperature-sensitive Gal80 transgene to restrict GFP degradation to a short window. Larvae were shifted from 18˚C (Gal80 active) to 30˚C (Gal80 inactive) to allow for GFP degradation 24hr prior to dissection. (B) (Left) smRNA FISH against the first intron of *Ubx* in *GFPUbx* fusion knockin haltere discs. (Middle) Cropped image of smRNA FISH from left. (Right) Acquisition of GFP fluorescence. "No Degrade" signifies the absence of a Gal4 to drive the degrade system. "Distal Degrade" denotes the use of *nubbin-Gal4* (*nub-Gal4*), and "Proximal Degrade" denotes the use of *teashirt-Gal4* (*tsh-Gal4*). (C) The same experiment as in B for the *GFP-t2a-Ubx* allele. (D) A model of Ubx negative autoregulation in the proximal haltere. Multiple low affinity Ubx-Exd binding sites within *abxFAIRE* are necessary for proximal repression, but we cannot rule out the involvement of additional sequence juxtaposed to *abxFAIRE* or elsewhere in the gene. Because *abxFAIRE* is necessary for the activation of *Ubx* expression (both distally and proximally), we suggest the possibility that the proximal repression effect results from a modification of this inherent activating role. All scale bars are 50 micron in size.

phenomenon is evolutionarily conserved. In addition, haltere development in flies is very sensitive to *Ubx* dose, which likely reflects the requirement for precise Ubx levels [13,16]. More

generally, because stereotyped differences in transcription factor expression levels *in vivo* is a widespread phenomenon, the mechanisms described here are likely to be relevant to many biological systems.

By performing transgenic *abx* reporter assays and genome engineering of the native *abx* CRM in parallel, we were able to reveal disparities in the impact of binding site mutations in each context. While we established the role of Ubx (through targeted protein degradation) and Hth-Exd (through the use of genetic null alleles) in mediating proximal repression at the endogenous locus, mutation of 14 predicted binding sites within the native *abx* CRM did not result in de-repression of Ubx, despite having this effect in a transgenic reporter context. This disparity, along with our understanding of the involvement of multiple low affinity binding sites throughout the *abx* region, suggests a model in which proximal repression and the formation/maintenance of the PD bias relies on the contribution of sequence information both within and outside the *abx-FAIRE* region (Fig 7D). We focused on a ~4 kb region within *abx* that both exhibits the highest signal in FAIRE accessibility assays, shows a significant amount of Ubx and Hth binding, and is able to recapitulate the PD bias when paired with a reporter. Despite this, additional regions of chromatin accessibility and Ubx-Hth binding are observed within the locus, and may also contribute to PD bias formation (Fig 3A). Consistent with this notion, the 5' UCR, as defined by a 35 kb region upstream of the *Ubx* TSS, also drives PD-biased expression [13]. Thus, the *Ubx* PD bias is not established by a single regulatory element, but instead is likely the consequence of a more dispersed activity that down-regulates the levels of many (or all) CRMs within the locus.

Also relevant to this idea is that when the endogenous Cluster 1 binding site was mutated to be higher affinity, we failed to detect an effect on proximal *Ubx* expression levels. Although we anticipated that this gain-of-function mutation would increase recruitment of Ubx to the locus, resulting in an even greater proximal repression as observed in the transgene context, the lack of effect may be a consequence of the mutation of a single cluster in the context of many Ubx binding sites. It may be that this mutation does not significantly impact the combined affinity of all binding sites within the region, resulting in a relatively normal amount of Ubx occupancy at the endogenous *abx* region. Further, a recent report, which found that multi-enhancer "hubs" at the Ubx-responsive *shavenbaby* locus can buffer against environmental stress and genetic perturbation [39], suggests that the entire 3D environment in which *abx* is localized may also be relevant to how *Ubx* levels are buffered and why manipulating a small number of binding sites failed to produce an obvious change in Ubx levels. Future studies, which either manipulate hub formation or that systematically increase the number and affinity of multiple Ubx binding sites in *abx*, may provide additional evidence to support this model.

Given that *abx* mediates both activation of *Ubx* transcription (proximally and distally) and down-regulation of *Ubx* transcription (proximally), it will also be interesting to determine if Ubx-mediated down-regulation of *Ubx* transcription occurs by either blocking the inherent activity of *abx* or, alternatively, by altering *abx*'s ability to engage with the *Ubx* promoter. Further, we note that *Ubx* down-regulation does not occur in all cells that express Ubx, Hth, and Exd, such as in distal hinge cells, implying that additional factors (either proximally or distally) are likely to modulate the autorepressive activity of Ubx at *abx*.

## Methods

### Fly strains

All wildtype strains used are *yw*. Mutant alleles used are as follows: *exd¹*, *exd²*, *hth^P2^*, *hth^100-1^*, *Ubx^9-22^*, *Dcr²*. UAS-*ExdRNAi* was obtained from the Vienna Stock Center (VDRC), and UAS-*UbxRNAi* was generated in our lab using the following primer sets–UbxF-NheI: GATCGCTA GCAACTCGTACTTTGAACAGGCC; Ubx-F-AvrII: GATCCCTAGGAACTCGTACTTTGA

ACAGGCC; Ubx-R-XbaI: GATCTCTAGAGCTGACACTCACATTACCGC; Ubx-R-EcoRI: GATCGAATTCGCTGACACTCACATTACCGC. *Ubx-lacZ$^{HC166D}$* (also referred to as *Ubx-lacz$^{166}$* [16]) was a gift from Welcome Bender and is described in Bender *et al.* 2000 [23]. UAS-GFPdegrade flies (UAS-NSlmb-vhhGFP4) are from Bloomington [34]. The following CRISPR alleles (described below) were made for and used in this study: Targeted Region$^{2kb-ubiDsRed}$ Replacement Platform, *abxN$^{WT}$*, *abxN$^{Cluster1Kill}$*, *abxN$^{Cluster1Hi}$*, *Targeted RegioN$^{4kb-ubiDsRed}$* Replacement Platform, *abxFAIRE$^{WT}$*, Δ*abxFAIRE* (which is: abxFAIRE$^{MCS}$), *abxFAIRE$^{Set2KILL}$*, *abxFAIRE$^{Set1KILL}$*, , *abxFAIRE$^{Set1KILL+Set2KILL}$* abxFAIRE-abxN$^{WTreplace}$, abxFAIRE-abxN$^{Cluster1Killreplace}$, *UbxExon1$^{P3RFP}$* Replacement Platform, *UbxExon1$^{WT}$*, *GFPUbx* Fusion, *GFP-t2a-Ubx*. Each allele was validated with Southern Blot and PCR and characterized. Deletion of *abxFAIRE* is generally lethal, with the rare survival of homozygous viable adults, which exhibit an intermediate haltere-to-wing transformation. Replacements of *abxFAIRE* with the smaller *abxN* allows for the survival of viable homozygous adults, which also exhibit intermediate haltere-to-wing transformations.

## CRISPR targeting

Two regions within *abx* and one region encompassing *Ubx* Exon1 were targeted with CRISPR/Cas9. For each targeting event, two gRNAs were designed flanking the region of interest. gRNA sequences for each of the regions are as follows–*abx-Targeted Region$^{2kb}$*: GAGATGCTTTTGAATTCTCG and GGCAGATCGGATTGGATCTT; *abx-Targeted Region$^{4kb}$*: GGCTTTGCAACTAATTGAAA and GTAAATGTTGGCTATTCAAAA; *Ubx* Exon1: GAATTCGAAGAAAATTAG and GTAAGACATATGAAAGC. gRNAs were cloned into the pCFD4 dual gRNA vector (http://www.crisprflydesign.org/, Port *et al.* 2014 [40]). Homemade donor vectors were made containing either a ubiDsRED or P3-RFP fluorescent selection marker flanked by inverted PhiC31 attP recognition sequences–the ubiDsRED cassette contains a full attP sequence (GTACTGACGGACACACCGAAGCCCCGGCGGCAACCCTCAGCGGATGCCCCGGGGCTTCACGTTTTCCCAGGTCAGAAGCGGTTTTCGGGAGTAGTGCCCCAACTGGGGTAACCTTTGAGTTCTCTCAGTTGGGGGCGTAGGGTCGCCGACATGACACAAGGGGTTGTGACCGGGGTGGACACGTACGCGGGTGCTTACGACCGTCAGTCGCGCGAGCGCGA), whereas the P3-RFP cassette contains a minimal attP sequence (CCCCAACTGGGGTAACCTTTGAGTTCTCTCAGTTGGGGG) from Voutev *et al.* 2018 [41]. ~1.5 kb homology arms were cloned on either side of the inverted attP sites. Primers used to clone the homology arms are as follows: *abx-Targeted Region$^{2kb}$*: (Left Arm) GCCAGAAGCTGCAAATTCAAG and CTTTGGGTTCTGTTCCACAGC, (Right Arm) GAATTCAAAAGCATCTCCGCATAAAG and GCCAACCGCAGACTGTGCGA; *abx-Targeted Region$^{4kb}$*: (Left Arm) GATGTAGGCCATGGTTTCGGC and TGAATAGCCAACATTTACTGACTCG, (Right Arm) AAACGGTAAAACTTGAGATTTTCTTATT and CGGAGAATCCGTATGAATCG; *Ubx Exon1*: (Left Arm) GCTCAACTGTAGTTTTCTGTTCG and ATTTTCTTCGAATTCTTATATGCTAT, (Right Arm) AGCAGGCAGAACAGACCTT and CTCGCAGAGATTGTCTGACAC. The gRNA template (pCFD4) and donor template were injected into a germline-expressing Cas9 strain (nanos-Cas9, Kondo *et al.* 2013 [42]) at a concentration of 250 ng/μL and 500 ng/μL, respectively. Selection of positive CRISPR events was done by screening for the presence of ubiDsRED or P3-RFP. Positive fly lines were validated by PCR and Southern Blot analysis.

## Recombinase mediated cassette exchange (RMCE)

PhiC31-mediated RMCE was used to replace the ubiDsRED/P3-RFP selection markers inserted using CRISPR/Cas9 into *abx/Ubx Exon1*. A homemade vector was used, containing inverted PhiC31 attB recognition sequences (CGGGTGCCAGGGCGTGCCCTTGGGCTCC

CCGGGCGCGTAC) flanking a multiple cloning site for insertion of sequences used for replacement alleles. Replacement of wildtype sequence from *abx-Targeted Region²ᵏᵇ*, *abx-Targeted Region⁴ᵏᵇ*, and *Ubx* Exon1 was performed by amplifying regions of interest from the yw genome. The following primers were used–*abx-Targeted Region²ᵏᵇ*: ATCCAATCCGATCT GCCCAG and TCGAGGAGTGAGTAAGAGATTGATAAAG; *abx-Targeted Region⁴ᵏᵇ*: AAACGGGAGGCTTTTGCTG and CAATTAGTTGCAAAGCCGTTTTTC; *Ubx Exon 1*: TAGAGGTTGTATTGTTTTATTAATAAAAAACCTATTG and TTCATATGTCTTACAT TACAAGTTGTTATCTGTTTTTCC. Mutations of replacement regions were made either by site directed mutagenesis or through synthesis and ligation of mutated restriction fragments within the region of interest. In the generation of *abx-Targeted Region⁴ᵏᵇ* mutant replacements, the ~4 kb fragment with mutations was stitched together from synthesizing restriction fragments based on reference sequence with engineered mutations. In doing this, additional natural variants (SNPs, small indels) between the reference strain and our wildtype strain (yw) were introduced into the *abxFAIRE*ᴿᴱᵗ²ᴷᴵᴸᴸ and *abxFAIRE*ˢᵉᵗ¹ᴷⁱˡˡ⁺ˢᵉᵗ²ᴷⁱˡˡ, and thus differ from the wildtype *abxFAIRE*ᵂᵀ replacement allele in these additional locations. These natural variants do not reside within predicted Ubx/Exd binding sites, do not create new predicted Ubx/Exd binding sites, nor do our results show that they have an effect on PD bias formation. The *abxFAIRE* deletion allele was generated by using an attB donor plasmid containing a multiple cloning site (gaagcttcctaggaggcctagatctgcggccgcttaattaaacgcgtgaatgggcgcgccgctagccatatggg-taccggatcc). The replacement of *abxFAIRE* with *abxN* was done using the following primers: GAACACAAAGGAGTCTGGTG and AACGTCGGAGGATGTAGG. The GFPUbx fusion and GFP-t2a-Ubx replacement alleles were cloned using overlap PCR and inserted at the transcription start site of *Ubx Exon 1*. The GFPUbx fusion contained the linker, GGSGGSG; the GFP-t2a-Ubx allele contained the t2a sequence, ggttctggagagggccgcggcagcctgctgacctgcggc-gatgtggaggagaaccccgggccc. Replacement donor plasmids were injected into flies with the attP platform cassette either at *abx* or at *Ubx Exon 1*. The necessary recombinase enzyme, PhiC31, was either injected as plasmid along with the donor cassette (*abx* replacements) or was expressed from a genomic insertion on the X chromosome of nanos-PhiC31 (*Ubx Exon 1* replacements). Progeny from injected flies were screened for the loss of the fluorescent selection marker (UbiDsRED, P3-RFP). Because the attP/attB reaction does not provide directionality, replacements can be inserted in the forward or reverse direction. Southern blot was performed to ensure the correct directionality of the replacement.

## Antibodies

Antibodies used are as follows–anti-Ubx (mouse, 1:10 FP3.38 from Developmental Studies Hybridoma Bank (DSHB) in supernatant or ascites form); anti-ß-gal (Rabbit, 1:5000, MP Biochemicals 559762); anti-Hthᴴᴰ (guinea pig, 1:500, Noro and Mann 2006 [26]); anti-Hth (guinea pig, 1: 5000, Ryoo and Mann 1999 [43]); anti-Exd (rabbit, Mann and Abu-Shaar 1996 [44]); where GFP signal is shown, native GFP fluorescence was acquired.

## FAIRE and ChIP-chip data accession

FAIRE data shown was from [28], and downloaded from NCBI GEO database with accession number: GSE38727. ChIP data for Ubx and Hth was obtained from Slattery *et al.* [29], and downloaded from NCBI GEO database with accession number: GSE26793.

## Immunostaining of imaginal discs

Wandering third instar larvae were collected and dissected in PBS to invert the head region and expose attached imaginal discs to solution. Inverted heads were fixed in Fix Solution

(PBS/4% Paraformaldehyde/.1% TritonX/.1% Sodium Deoxycholate) for 25 minutes at RT. Fix solution was removed and replaced with Staining Solution (PBS/.3% TritonX/1% BSA). Inverted heads were washed 2X with Staining Solution for 20 minutes at RT and stained overnight with desired antibody in Staining Solution at 4C. The following day antibody solution was removed and inverted heads washed 4X with Staining Solution followed by a 1.5hr incubation with secondary antibody and DAPI (1:1000) in Staining Solution at RT. This was followed by two washes with Staining Solution, dissection of discs from the inverted heads in PBS and mounting of the discs in Vectashield. Imaging of discs was conducted on the following microscopes: Zeiss Apotome.2 Microscope, Leica SP5 Confocal Microscope, and Zeiss LSM 800 Confocal Microscope.

## Calculation of distal:Proximal ratio

All analysis was done in Fiji/ImageJ. Confocal Z-stacks of discs were imported into Fiji and slices near the edge of the stack that contained peripodial membrane were manually removed. All pixels surrounding the haltere disc of interest were cleared by drawing an ROI around the disc itself. ROIs for the distal compartment and proximal compartment were drawn based on a single Z-slice in which the distal pouch was clearly demarcated (using either the antibody stain of interest (Ubx/LacZ) or DAPI). These ROIs were propagated to all slices of the image and the average intensity of the stain of interest was acquired, excluding black pixels. For each disc, the mean of these single-slice average intensities was computed for the whole disc, distal compartment, and proximal compartment. Multiple discs were analyzed to produce the scatter plots shown. Each point is representative of a single disc. A one-way ANOVA analysis followed by Tukey's multiple comparisons test with alpha = .05 was used for statistical analysis. This process is depicted in a schematic in S1 Fig.

## Imaging of adult haltere structure

Whole adult flies were submerged in 70% ethanol, followed by three washes in PBS/.3% TritonX. The head and abdomen were removed, leaving the thorax with appendages attached. Thoraces were fixed overnight at 4˚C in PBS/4% Paraformaldehyde. The following day, thoraces were washed 5X with PBS/.3% TritonX, followed by dissection and mounting of halteres in Vectashield. Imaging was conducted on a Leica SP5 confocal microscope, acquiring autofluorescence of the cuticle with the 488 laser. Images shown are MAX projections of the stacks acquired.

## Generation of lacZ reporter constructs

All CRM lacZ reporter constructs were generated by cloning the regulatory DNA of interest (*abxN*, *abxF*) into pRVV54-lacZ [45] using the NotI and HindIII restriction enzyme sites. Primers (5' to 3') used to amplify each regulatory region are as follows–*abxN*: GAACACAAA GGAGTCTGGTG and AACGTCGGAGGATGTAGG; *abxF*: GAACACAAAGGAGTCTG GTGAG and GTTAAGCATTTTGGGTGCGAG. All lacZ reporter constructs were inserted into the attp40 landing site on chr2. Mutations were made either through site-directed mutagenesis or through synthesis of mutated restriction fragments within the regulatory region of interest, which were then ligated back into the pRVV54-lacZ backbone.

## NRLB binding site prediction

Binding sites for Exd/UbxIVa were predicted using the NRLBtools package in R [30].

### Generation of RNAi flip-out clones

Flip-out clones were generated by crossing *act<y<Gal4, UAS-GFP* to different *hs-flp*; *UAS* lines and heat-shocking larvae for between 8–10 minutes at 37C.

### Generation of mitotic clones

Mutant alleles of interest were recombined with standard FRT lines. Flies with mutant recombined alleles were crossed to either FRT ubiGFP or ubiDsRED (to mark the clones) and progeny of this cross were heatshocked at 37C for 40min-1hr, 48 hours after egg laying (AEL). Wandering third instar larvae were collected 72 hours after heatshock, dissected, and subjected to the immunostain protocol as described.

### ChIP-qPCR

Wandering third instar larvae from two different genotypes (*abxN WT-lacZ* and *abxN Cluster 1$^{kill}$*-lacZ) were dissected and haltere imaginal discs were collected in PBS on ice. Discs were fixed with 1.8% formaldehyde, crosslinked chromatin was sonicated, and chromatin preparation and immunoprecipitation was performed as described in Estella *et al*. 2008 [46]. The IP was done with rabbit anti-Ubx (Ubx1, generated by modENCODE) at a final concentration of 1.5 µg/mL for each IP. Rabbit IgG (Sigma) was used for the control IP. The following primer pairs were used for qPCR–Endogenous *abx*: TGGAGCTCCAAATGAAACGC and CGCTC AACATTGTTAGTGGC; Transgenic *abxN*: CAGTGCTGGCTGCATTTGCT and ACAACT GATGCTCTCAGCCA; Intergenic Control: CCGAACATGAGAGATGGAAAA and AAAGT GCCGACAATGCAGTTA. qPCR was done on an Applied Biosystems 7300 machine and calculations were done using the 2-ΔΔCt method in MS Excel. IPs were done in triplicate.

### Protein purification and electrophoretic mobility shift assays (EMSAs)

Ubx protein was His-tagged and purified from *E. coli* (BL21 or BL21pLysS; Agilent) 4 hours of induction with isopropyl-B-D-thiogalactopyranoside (IPTG) using Co-chromatography. Exd (in pET9a) and Hth$^{HM}$ (in pET21b) were co-expressed and co-purified, through the His-tag attached to Hth$^{HM}$, in *E. coli* (BL21) in the same way as Ubx recombinant protein and used as a complex for all EMSAs. Protein concentrations were determined by the Bradford assay and then confirmed by SDS/PAGE and Blue Coomassie analysis (SimplyBlue SafeStain, Invitrogen). EMSAs were carried out as previously described ([47]). UbxIa was used at 250 ng/lane (high concentration) and 200 ng/lane (low concentration) and Exd/HthHM was used at 150 ng/lane. Sequences for probes used are as follows–*abxWT*: CCTCGTCCCACAGCTcgaatatat tttataacccggagcaaatgcagcca; *abxCluster1$^{Kill}$*: CCTCGTCCCACAGCTcgaatccccttccccacccggagc aaatgcagcca. Uppercase is linker sequence. DNA binding was observed using phosphoimaging as detected by a Typhoon Scanner (Amersham).

### smRNA FISH

A probe library containing 48 20-nt Stellaris FISH probes was designed to target the first 2 kb of Ubx Intron 1. Libraries were ordered from Biosearch Technologies and labelled with Quasar 670. Wandering third instar larvae were collected and dissected in PBS to invert heads and expose discs to solution. Inverted heads were washed in PBSM (PBS/5mM MgCl2) 1X at RT, followed by fixation in PBSM/4% PFA for 10 min at RT. Discs were permeabilized with PBS/ .5% TritonX for 10 min at RT and washed once with PBSM for 10 min at RT. Inverted heads were washed 1X with Pre-Hyb (10% deionized formamide in 2X SSC) for 10 min at RT prior to hybridization. Hybridization was performed overnight in a thermoshaker at 37˚C (~600RPM)

covered in foil. Hybridization buffer contains: 2X SSC, .2 mg/mL BSA, 50% Dextran Sulfate, 10% deionized formamide, 50 µg/mL *E. coli* tRNA, 50 µg/mL salmon sperm ssDNA, and 125 nM Ubx Intron Probe. 100 µL of hybridization buffer was used for each sample. The following day, hybridization buffer was removed and heads were washed with Pre-Hyb buffer for 20 minutes at 37˚C and 20 min at RT. Inverted heads were washed with PBS for 10 min at RT, stained with DAPI (PBS/DAPI (1:1000 dilution) for 30 min at RT and resuspended in PBS. Discs were dissected from inverted heads in PBS/1%BSA and mounted in Vectashield prior to imaging.

### GFP degrade assay

deGRADFP flies, UAS-NSlmb-vhhGFP4, were obtained from Caussinus *et al*. 2011 [34] and paired with tubGal80ts, and either nubbin (nub)-Gal4 or teashirt (tsh)-Gal4. Flies were crossed to GFPUbx fusion or GFP-t2a-Ubx knockin flies. Progeny were kept at 18˚C (Gal80 inactive) until early third instar stage and then shifted to 30˚C for 24 hours prior to dissection. Wandering third instar larvae were collected, dissected, and subjected to smRNA FISH analysis. Genotypes are as follows: "No Degrade:" yw; UAS-GFPdegrade, gal80ts/UAS-GFPdegrade, gal80ts; GFPUbx (or GFP-t2a-Ubx)/GFPUbx (or GFP-t2a-Ubx); "Distal Degrade:" yw; nubGal4/ gal80ts, UAS-GFPdegrade; GFPUbx (or GFP-t2a-Ubx)/ GFPUbx (or GFP-t2a-Ubx); "Proximal Degrade:" yw; tshGal4/gal80ts, UAS-GFPdegrade; GFPUbx (or GFP-t2a-Ubx)/ GFPUbx (or GFP-t2a-Ubx).

## Supporting information

**S1 Fig. Overview of strategy to quantify the proximal/distal bias of Ubx expression. (A)** Analysis pipeline conducted within FIJI is shown. Selection of whole disc region of interest (ROI), Distal ROI, and Proximal ROI is followed by the measurement of the average intensity of Ubx within that ROI for each slice in the stack. Black pixels are removed from the analysis to prevent skewing of the average from differences in sizes of the ROI. **(B)** (Left) An analysis of a single haltere disc is shown. Each dot represents the average intensity for each slice in the stack in the whole disc ("Disc"), the proximal ROI, and the distal ROI. Dotted line represents the mean for each compartment; reported distal:proximal ratio is the distal mean/proximal mean. **(C)** As shown in Fig 1. The mean average Ubx intensity for each compartment from the single disc analysis is reported as a single point in the multi-disc analysis. Individual discs are color-coded such that the compartment-specific averages for each disc can be compared to one another. Dotted line is the mean average intensity for each compartment.
(TIF)

**S2 Fig. Additional *exd* mutant clones. (A)** Ubx Immunostain in haltere discs in which *exd* null clones (*exd²*) have been induced. An Exd immunostain, in addition to a merge of Ubx/ Exd, is shown. Clones are marked by a yellow arrow. **(B)** Ubx immunostain in two haltere discs in which *exdRNAi* clones have been induced in the background of a *Dcr2* mutant. An Exd immunostain, in addition to a merge of Ubx/Exd, is shown. Clones are marked by a yellow arrow.
(TIF)

**S3 Fig. Clonal deletion of *abxFAIRE*. (A)** (Top) A schematic of the *abxFAIRE* region targeted and deleted. FAIRE accessibility peaks [28] are shown in the wing and haltere. (Bottom) The Δ*abxFAIRE* allele was generated by replacing the *abxFAIRE* sequence with a minimal cloning site (MCS) sequence using PhiC31-based RMCE (see methods). Through this method, scars are left on either side of the replacement. **(B)** Mitotic clones homozygous for Δ*abxFAIRE* (GFP-) were induced at 48hr after egg-laying (AEL). Clones in the posterior compartment

(*EnGal4*-RFP+) do not show a defect in Ubx expression. Clone is outlined in yellow. **(B')** A zoomed in image of the clone from **B**. Merges of Ubx/GFP and *EnGal4*-RFP/GFP are shown. Scale bars are 50 micron in size.
(TIF)

**S4 Fig. Loss of Cluster 1 binding sites results in loss of Ubx binding. (A)** (Left) LacZ immunostain of haltere discs containing either an *abxN-lacZ*$^{WT}$ transgene or an *abxN-lacZ*$^{Cluster1-Kill}$ transgene. (Right) LacZ immunostain of discs of wildtype and mutant *abxN-lacZ* genotype upon induction of *Ubx* null clones (*Ubx*$^{9-22}$). A GFP immunostain, in addition to a merge of GFP/LacZ, is shown. Clones are GFP-negative. **(B)** EMSA assay of Ubx-Exd-Hth$^{HM}$ *in vitro* binding to a probe containing the Cluster 1 binding sites (left) and the probe mutated to abrogate the Cluster 1 binding sites (right). Gray arrow points to the shifted trimer band. **(C)** Chromatin Immunoprecipitation (ChIP-qPCR) was performed on haltere discs from transgenic flies containing either the *abxN-lacZ*$^{WT}$ reporter (left) or the *abxN-lacZ*$^{Cluster1-Kill}$ reporter (right). % Input for an anti-Ubx IP and an IgG Isotype control IP are shown for three genomic regions: (1) the endogenous *abx* CRM, (2) the transgenic CRM, and (3) an intergenic region on chromosome 2 that serves as a negative control. Averages and standard deviation from three independent IPs are reported. **(D)** Quantification of raw LacZ intensities in discs from a single experiment out of a total of three, which were reported together in Fig 4C and 4C'. Values are reported for the average LacZ intensity of the whole disc, the distal compartment, and the proximal compartment. Each dot represents a single disc, and the dotted line signifies the mean value. While the trend in the data is representative of that in Fig 4C, none of the differences here are statistically significant (ANOVA, cutoff .05), likely because of the large amount of variability in these measurements and small sample size of a single experiment. **(D')** Quantification of normalized LacZ intensities from **D.** Proximal and Distal LacZ average intensities for each disc are divided by the average intensity of the whole disc. This quantification provides a measurement of the contribution of proximal and distal LacZ levels to the average of the whole disc. An inverse relationship exists between proximal and distal levels, such that whole disc levels remain approximately the same upon mutation (**D, left**) but distal and proximal levels change in opposition to one another. Each point represents a single disc, and the dotted line signifies the mean value.
(TIF)

**S5 Fig. Generation of clones homozygous for wildtype and mutant *abxN* replacement alleles. (A)** GFP and Ubx immunostains in haltere discs in which clones homozygous for *abxN* replacement alleles were induced 48hr AEL. Clones are GFP-negative and denoted with a yellow arrow. Zoomed images of single clones (outlined) are shown to the right. All scale bars shown are 50 micron in size.
(TIF)

**S6 Fig. Multiple low affinity Ubx/Exd binding sites are required for PD bias formation. (A)** Table of *NRLB*-predicted Ubx/Exd binding sites in *abxFAIRE* that have been mutated. Many predicted sites fall within clusters so we report the number of binding sites within each cluster. Wildtype and mutated sequences are shown. Lowercase levels are bases that were mutated. Relative affinities and the category (Set1, Set2) for each binding site or cluster are given. **(B)** GFP and Ubx immunostains in haltere discs in which clones homozygous for *abxFAIRE* replacement alleles were induced 48hr AEL. Clones are GFP-negative and denoted with a yellow arrow. Zoomed images of single clones (outlined) are shown to the right. All scale bars shown are 50 micron in size.
(TIF)

**S7 Fig. Multiple low affinity Ubx/Exd binding sites are required for PD bias formation. (A)** Schematic of *abxFAIRE* replacements tested in **B**. The Targeted Region[4kb] replacement platform was used to generate either a wildtype *abxFAIRE* (+scars) allele or an allele with only the ~500 bp *abxN* wildtype or Cluster 1[kill] sequence, deleting the remaining ~3.5 kb *abxFAIRE* sequence. **(B)** Ubx immunostain in haltere discs homozygous for the specified *abxFAIRE* replacement alleles. A DAPI nuclear stain is shown for each. All scale bars shown are 50 micron in size.
(TIF)

**S8 Fig. CRISPR targeting of *Ubx* exon 1. (A)** Genome browser screenshot around *Ubx* exon 1. FAIRE accessibility peaks in the wing and haltere [28] are shown for reference. Conservation track was downloaded from UCSC. No bars denote a lack of evolutionary conservation with twelve other *Drosophila* species, mosquito, honeybee and red flour beetle. Yellow triangles denote Cas9/gRNA target sites; and the yellow bar denotes the region replaced. **(B)** Schematic of a two-step *Ubx* exon 1 replacement strategy. gRNA sites were chosen in non-conserved regions surrounding *Ubx* exon 1. A dual-gRNA expressing plasmid was injected along with a donor cassette containing an attP-flanked fluorescent selection marker (*P3-RFP*) into *nanos-Cas9* flies. The resulting "*Ubx* Exon 1 Replacement Platform" serves as a means to insert modified versions of *Ubx* using PhiC31-based RMCE. **(C)** (Left) Schematic of RMCE replacement cassette. (Right) Schematic of replacement alleles generated and sequence information for the linker used in the GFP-Ubx fusion (amino acids) and the T2A sequence used (DNA). **(D)** GFP native fluorescence and a Ubx immunostain in haltere discs in which clones homozygous for the wildtype *Ubx* exon 1 replacement allele were induced. Clones are GFP-negative and marked with a yellow arrow. Cropped images of single clones (outlined) are shown. **(E)** GFPUbx native fluorescence, and dsRed and Ubx immunostains in haltere discs in which clones homozygous for the GFPUbx fusion allele were induced. Clones are dsRed-negative and marked with a yellow arrow. Cropped images of a single clone (outlined) are shown. **(F)** GFP native fluorescence, and dsRed and Ubx immunostains in haltere discs in which clones homozygous for the *GFP-t2a-Ubx* allele were induced. Clones are dsRed-negative and marked with a yellow arrow. Cropped images of a single clone (outlined) are shown. **(G)** An anti-Ubx immunoblot on protein derived from the following genotypes: *yw* (lane 1), *GFP-Ubx* heterozygous (lanes 2, 3), and *GFP-t2a-Ubx* (lane 4). The lower band is Ubx and the upper band is the GFPUbx fusion. All scale bars shown are 50 micron in size.
(TIF)

## Acknowledgments

The authors thank members of the Mann lab for comments and suggestions throughout this study and Chaitanya Rastogi and Harmen Bussemaker for help with binding site analyses.

## Author Contributions

**Conceptualization:** Rebecca K. Delker, Vikram Ranade, Richard S. Mann.

**Formal analysis:** Rebecca K. Delker, Vikram Ranade, Richard S. Mann.

**Investigation:** Rebecca K. Delker, Vikram Ranade, Ryan Loker, Roumen Voutev.

**Methodology:** Rebecca K. Delker, Vikram Ranade, Ryan Loker, Roumen Voutev.

**Project administration:** Richard S. Mann.

**Supervision:** Richard S. Mann.

**Visualization:** Rebecca K. Delker, Vikram Ranade, Richard S. Mann.

**Writing – original draft:** Rebecca K. Delker, Vikram Ranade, Richard S. Mann.

**Writing – review & editing:** Rebecca K. Delker, Richard S. Mann.

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
