## [Decision Letter · Decision Letter 0]

26 Aug 2019

[EXSCINDED]

Dear Dr Mann,

Thank you very much for submitting your Research Article entitled 'Low Affinity Binding Sites in an Activating CRM Mediate Negative Autoregulation of the Drosophila Hox Gene Ultrabithorax' to PLOS Genetics. Your manuscript was fully evaluated at the editorial level and by independent peer reviewers. The reviewers appreciated the attention to an important problem in gene regulation and the powerful combination of different approaches used to examine negative autoregulation of Ubx. There is interest in publishing this work but the reviewers raised some important concerns about the relevance and interpretation of findings as presented in the current manuscript. Based on the reviews, we will not be able to accept this version of the manuscript, but we would be willing to review again a revised version that addresses and/or comments upon these concerns. We cannot, of course, promise publication at that time.

With respect to revisions, reviewers 1 and 2 are very supportive of publication. Reviewer 1 has minor concerns that should be easily addressed. Their suggestion of noting the idea that " transcriptional hubs"  might buffer deletions in the low-affinity binding sites in the CRM seems sensible. Reviewer 2 raised a major concern about the relevance or in vivo role of the P-D gradient in the haltere. This needs to be discussed. They also raised a series of interesting and relevant points (minor points 1-6) , including whether Ubx is required for the negative autoregulation. They noted that in the wing disc the lacZ reporter is also expressed in a P-D gradient in the absence of Ubx. They suggest this might imply Antp plays a role in that context or that Hth and Exd function on this element in haltere and wing discs without input from Ubx. These points need to be considered and addressed. Reviewer 3 questions the important of the work because no new mechanisms are uncovered. However, I believe that this concern may be adequately addressed in response to the major and minor points  raised by reviewer 2.   

Should you decide to revise the manuscript for further consideration here, your revisions should address or rebut each of the specific points made by the reviewers. We will also require a detailed list of your responses to the review comments and a description of the changes you have made in the manuscript.

If you decide to revise the manuscript for further consideration at PLOS Genetics, please aim to resubmit within the next 60 days, unless it will take extra time to address the concerns of the reviewers, in which case we would appreciate an expected resubmission date by email to plosgenetics@plos.org.

[LINK]

We are sorry that we cannot be more positive about your manuscript at this stage. Please do not hesitate to contact us if you have any concerns or questions.

Yours sincerely,

Robb Krumlauf

Guest Editor

PLOS Genetics

Gregory Barsh

Editor-in-Chief

PLOS Genetics

Reviewer's Responses to Questions

**Comments to the Authors:**

Reviewer #1: In the manuscript “Low affinity binding sites in an activating CRM mediate negative autoregulation of the Drosophila Hox Gene Ultrabithorax,” Delker and colleagues explore new low-affinity Ubx binding sites in a ‘classic’ abx enhancer. Specifically, they identify and molecularly dissect the contribution of low affinity Ubx binding sites in the abx enhancer—an enhancer that contributes to a negative auto-regulatory loop that maintains low expression in a specific zone of the haltere disc. Intriguingly, they find that mutations sufficient to result in de-repression of abx activity in a transgene are not sufficient to de-repress in the endogenous locus. Together, these results nicely highlight the importance of low-affinity transcription factor binding sites as well as the need to understand gene regulation within the native context.

Overall, this is an excellent paper that is rigorous and timely. The authors characterize new Ubx binding sites in a classic enhancer, contributing to our understanding of how low-affinity binding sites shape animal development.

A few points may help the manuscript:

-ON/OFF: The distinction of ON versus OFF, i.e., binary expression, is not novel and detracts a bit from the intro. It feels like a “straw man” argument.

-Throughout, Red/Green figures should be changed for colorblind readers

- Figure 4: Could the authors show the predicted affinity changes from their mutations?

-Figure 4, 5, 6, etc.: “hi” should be change to “high.”

-Note: Some figure panels are lower resolution: Fig. 1A; Fig. 3A; Fig. 4S1B.

-Fig 6: These are great experiments; it could benefit readers if a different lookup table (LUT) were used (“Fire”). For example, in panel C, the low affinity Mut looks similar to the Cluster2Kill.

-Did the authors consider experiments to test how robust the modified endogenous locus is? For, example growing at elevated temperatures (Frankel et al., 2010; Crocker et al., 2015)? Is there a subtle phenotype (Tsai et al., 2019)? For example, smFISH could be compared to Ubx protein to look at noisy expression (or precision). Was there any morphological phenotype? These seems like a fantastic genetic reagent, it would be great to see if there are any subtle changes.

Discussion: have the authors considered how “hubs” may buffer the native locus? It would be interesting to consider local repression hubs across this region. Such a mechanism could function to stabilize the gene-expression output even if some low affinity binding sites are deleted.

Reviewer #2: Review of: “Low Affinity Binding Sites in an Activating CRM Mediate Negative Autoregulation of the Drosophila Hox Gene Ultrabithorax”

Summary: The authors use a combination of clonal knockouts, transgenic reporter assays, and mutations to endogenous regulatory DNA to investigate the regulation of Ubx expression in the haltere disc where a clear low-proximal high-distal expression pattern exists. The authors show that this differential proximal-distal expression requires negative autoregulation by Ubx and its cofactors Hth and Exd in the proximal compartment of the haltere disc. Further, they identify a regulatory region that drives expression in the Ubx pattern, and is repressed by Ubx and its cofactors via a cluster of low affinity binding sites. Mutations to this regulatory module in the endogenous locus fail to de-repress proximal Ubx expression suggesting sequences outside of this region are important for negative autoregulation.

Overall, the writing of the paper is very clear, the claims made are well supported, and a concerted effort was made to address each question using multiple approaches. A clear strength of the paper is the use of CRISPR to both manipulate the endogenous regulatory locus as well as to tag the endogenous Ubx coding region and to assess/compare their impact to that of transgenic reporter assays. Moreover, the authors performed a very large number of studies removing and adding back various versions of the Ubx-abx regulatory element. Using this approach, allowed the authors to highlight differences observed between studying regulatory DNA in a transgenic reporter versus an endogenous context. Transgenic reporter assays permitted the interrogation of single regulatory elements in isolation and identified changes driven by mutations to single binding sites. In contrast, by targeting the endogenous locus, which is perhaps the most biologically relevant, was complicated by the presence of potential redundant regulatory regions outside of the interrogated locus. In sum, these findings will be of interest to those studying Hox factors and their ability to regulate gene expression and they highlight the complexity of studying regulatory elements within their native context.

However, the paper does have several weaknesses. First, while I agree with the authors that we know a more about the regulation of gene expression at the level of ON vs OFF and less about how appropriate levels are achieved – it is unclear if the difference in Ubx expression levels between proximal and distal regions of the haltere disc has any physiological consequence. Because manipulation of the endogenous abx locus failed to disrupt proximal-distal Ubx expression bias, important questions like the following cannot be addressed: Is the differential proximal-distal Ubx expression required for haltere development, and would de-repression of Ubx in the proximal compartment of the haltere disc result in a phenotype/defect? Hence, the biological relevance of studying Ubx mediated autoregulation in the proximal haltere disc has not be established. In addition, I have several other concerns/suggestions, most of which are relatively minor, in regards to the manuscript.

1) Taking the abx enhancer out of its locus and studying it as a transgene permitted the detailed analysis for the requirement of specific binding sites – revealing that mutating the low affinity Ubx/Exd sites flattened the P/D differences in levels and increasing the affinity of sites increased the P/D levels. Based on these findings and clonal mutant analysis for Ubx, Hth, and Exd, the authors argue that Ubx requires low affinity sites to establish the differences in P/D expression. However, from the pictures in Fig 3C – it appears that the wing disc shows a similar P to D ratio in expression and this imaginal disc does not express Ubx. Does this mean that additional Hox factors (such as Antp) can regulate P/D expression in the wing or is this a Hox/Exd independent regulation? Is Hth and/or Exd required for this P/D difference in the wing as well? Answering this question could reveal new insight into Hox specificity or that there is also a Hox independent mechanism that can impact differences in P/D levels.

2) The authors provide evidence based on the LacZ transgene that low affinity sites are required for a proper P/D ratio of expression as raising the affinity steepens the differences seen. But if this low affinity site is required for proper P/D ratios, shouldn’t changing the low affinity sites to high affinity sites also impact the P/D ratios when inserted into the endogenous locus? In other words, while I understand and agree with the authors argument that the lack in change of P/D ratios when the endogenous sites were mutated could be due to redundancy via other elements with low affinity sites, I would have predicted that changing the low affinity sites to high affinity would have been dominant – and now one would need lower levels of Ubx to shut itself off and thus a steeper P/D ratio should be observed. The fact that no significant difference is observed calls into question whether the Ubx/Exd site just happens to be low affinity versus the Ubx/Exd site must be low affinity. This also ties into my above argument that unfortunately we do not know if the differences in P/D Ubx expression levels have a physiological consequence. Hence how important the sites must be low affinity is unclear.

3) Ubx only seems to negatively autoregulate its expression in the proximal compartment, and the same sites in the abxN element are repressed by Ubx proximally but not distally. Is the high-proximal low-distal expression of Hth and nuclear localization of Exd shown in figure 2 responsible? While it is clear these factors are required for proximal repression, would overexpression of Hth in the distal compartment using the nub-Gal4 presented in figure 7, result in distal repression?

4) Figure 4 shows that mutations to the predicted Ubx-Exd site in cluster 1 of abxN change proximal-distal expression bias decreasing the Distal to Proximal ratio when mutated to kill binding and increasing the ratio when the site is strengthened. However, it is less clear based on the images in 4C if this observation is due to changes in proximal expression levels (de-repression with Cluster1-kill and increased repression with Cluster1-high) or due to changes in distal expression levels. From the images shown it seems that distal expression levels are decreased in Cluster1-kill compared to WT which would decrease the D/P ratio, but would not support the hypothesis that the sites are required for repression in the proximal compartment. Could quantified proximal and distal levels be shown to indicate that not only does the D/P ratio change, but it is the result of de-repression/increased repression in the in the proximal haltere disc.

5) Minor: In Fig 4C – I believe the Y-axis should be labeled “Distal/Proximal bgal levels” and not “Distal/Proximal Ubx levels”.

6) Minor: I may have missed this and if I did the authors can ignore this comment, but is homozygous removal of the abx enhancer lethal? I assume since they are only showing clonal analysis, that it probably is, but it would be nice to state. Moreover, do any of the clonal enhancer mutants develop into adults and do they show haltere to wing transformations? Also, do the “patchy: AxbF and AxbN rescue lines” show haltere to wing transformations?

Reviewer #3: Review of Delker et al. for Plos Genetics 2019

In this paper, Delker and colleagues examine regulation of the Ubx gene with the goal of addressing on/off vs. analog gene expression. Ubx is off in the wing, but is on in the haltere. Interestingly, in the haltere, Ubx is expressed at different levels in different compartments – high in the distal, low in the proximal. They find that Ubx, hth, and exd negatively regulate Ubx expression in the proximal region. There is no mutual repression as Exd (levels and subcellular localization) are unaffected in Ubx clones. The authors characterize a previously identified CRM called abx. They generate enhancer reporters that recapitulate expression and regulation by Exd and Hth, Ubx, focusing on abxF reporter. Mutation of low affinity binding sites affects transgene expression but not endogenous expression. When they replace a large 4 kb region with a minimal ~500 bp sequence, they still observe Ubx autoregulation.

The authors convincingly show that Ubx, acting with Exd and Hth, negatively autoregulates. They also show that they can perturb this regulation in a minimal enhancer reporter. However, they did not identify DNA sequences whose mutation changes expression in the endogenous locus. The experiments are well-done, but ultimately, the story is incomplete. For publication in Plos Genetics, the authors would need to provide some insight into the negative feedback mechanism. For example, one simple hypothesis is that a shadow enhancer compensates for perturbation to abxF. If the authors could provide evidence for a shadow enhancer, this would greatly strengthen the paper. Alternatively, the authors could provide mechanistic insights with other types of experiments of their design.

Finally, the writing of the manuscript is dense and cryptic. The authors should edit the paper rigorously to make it more reader friendly. I list some of the passages that were extremely challenging to decipher:

“Importantly, we also reveal that Ubx-Exd-binding site mutations sufficient to result in de-repression of abx activity in the proximal haltere in a transgenic context are not sufficient to de-repress Ubx expression when mutated at the endogenous locus, suggesting the presence of multiple mechanisms through which Ubx-mediated repression occurs.”

“Though often described as the blueprint of an organism, the genome – and its underlying code – remains undecipherable unless understood within a cellular context. This is because, in addition to genomic sequence, knowledge of the subset of genes expressed throughout space and time is required.”

“While this demonstrates the importance of an ON/OFF binary mode of control for Ubx expression during development, it is also important to note that, although Ubx is detected in all cells of the haltere disc, distinct levels of Ubx expression occur along the proximal distal axis.”

“This observation raises questions not only of the downstream effects of high versus low Ubx expression on cell- and tissue-fate, but also of the mechanisms by which Ubx levels are regulated in proximal versus distal compartments. While the first question remains unanswered, here, we address the latter question and provide evidence for the presence of an autoregulatory mechanism in which Ubx directly represses its own expression within the proximal compartment of the developing haltere.”

**Have all data underlying the figures and results presented in the manuscript been provided?**

Reviewer #1: Yes

Reviewer #2: Yes

Reviewer #3: Yes

PLOS authors have the option to publish the peer review history of their article (what does this mean?). If published, this will include your full peer review and any attached files.

Reviewer #1: Yes: Justin Crocker

Reviewer #2: No

Reviewer #3: No

---

## [Editor Report · Decision Letter 1]

23 Sep 2019

Dear Dr Mann

We are pleased to inform you that your manuscript entitled "Low Affinity Binding Sites in an Activating CRM Mediate Negative Autoregulation of the Drosophila Hox Gene Ultrabithorax" has been editorially accepted for publication in PLOS Genetics. Congratulations!

Yours sincerely,

Robb Krumlauf

Guest Editor

PLOS Genetics

Gregory Barsh

Editor-in-Chief

PLOS Genetics

Comments from the reviewers (if applicable):

Thank you for submitting the revised version of the manuscript and dealing with the specific issues raised by the reviewers.

**Data Deposition**

http://datadryad.org/submit?journalID=pgenetics&manu=PGENETICS-D-19-01190R1

**Press Queries**

---

## [Editor Report · Acceptance letter]

3 Oct 2019

PGENETICS-D-19-01190R1 

Low Affinity Binding Sites in an Activating CRM Mediate Negative Autoregulation of the Drosophila Hox Gene Ultrabithorax 

Dear Dr Mann, 

We are pleased to inform you that your manuscript entitled "Low Affinity Binding Sites in an Activating CRM Mediate Negative Autoregulation of the Drosophila Hox Gene Ultrabithorax" has been formally accepted for publication in PLOS Genetics! Your manuscript is now with our production department and you will be notified of the publication date in due course.

With kind regards,

Kaitlin Butler

PLOS Genetics

On behalf of:
